# Lagging Inference Networks and Posterior Collapse in Variational Autoencoders

**Junxian He, Daniel Spokoyny, Graham Neubig**
Carnegie Mellon University
{junxianh,dspokoyn,gneubig}@cs.cmu.edu

**Taylor Berg-Kirkpatrick**
University of California San Diego
tberg@eng.ucsd.edu

## Abstract

The variational autoencoder (VAE) is a popular combination of deep latent variable model and accompanying variational learning technique. By using a neural inference network to approximate the model's posterior on latent variables, VAEs efficiently parameterize a lower bound on marginal data likelihood that can be optimized directly via gradient methods. In practice, however, VAE training often results in a degenerate local optimum known as "posterior collapse" where the model learns to ignore the latent variable and the approximate posterior mimics the prior. In this paper, we investigate posterior collapse from the perspective of training dynamics. We find that during the initial stages of training the inference network fails to approximate the model's true posterior, which is a moving target. As a result, the model is encouraged to ignore the latent encoding and posterior collapse occurs. Based on this observation, we propose an extremely simple modification to VAE training to reduce inference lag: depending on the model's current mutual information between latent variable and observation, we aggressively optimize the inference network before performing each model update. Despite introducing neither new model components nor significant complexity over basic VAE, our approach is able to avoid the problem of collapse that has plagued a large amount of previous work. Empirically, our approach outperforms strong autoregressive baselines on text and image benchmarks in terms of held-out likelihood, and is competitive with more complex techniques for avoiding collapse while being substantially faster.[1]

## 1 Introduction

Variational autoencoders (VAEs) (Kingma & Welling, 2014) represent a popular combination of a deep latent variable model (shown in Figure 1(a)) and an accompanying variational learning technique. The generative model in VAE defines a marginal distribution on observations, $\mathbf{x} \in \mathcal{X}$, as:

$$p_{\boldsymbol{\theta}}(\mathbf{x}) = \int p_{\boldsymbol{\theta}}(\mathbf{x}|\mathbf{z})p(\mathbf{z})\mathrm{d}\mathbf{z}. \qquad (1)$$

The model's generator defines $p_{\boldsymbol{\theta}}(\mathbf{x}|\mathbf{z})$ and is typically parameterized as a complex neural network. Standard training involves optimizing an evidence lower bound (ELBO) on the intractable marginal data likelihood (Eq.1), where an auxiliary variational distribution $q_{\boldsymbol{\phi}}(\mathbf{z}|\mathbf{x})$ is introduced to approximate the model posterior distribution $p_{\boldsymbol{\theta}}(\mathbf{z}|\mathbf{x})$. VAEs make this learning procedure highly scalable to large datasets by sharing parameters in the inference network to amortize inferential cost. This amortized approach contrasts with traditional variational techniques that have separate local variational parameters for every data point (Blei et al., 2003; Hoffman et al., 2013).

While successful on some datasets, prior work has found that VAE training often suffers from "posterior collapse", in which the model ignores the latent variable $\mathbf{z}$ (Bowman et al., 2016; Kingma et al., 2016; Chen et al., 2017). This phenomenon is more common when the generator $p_{\boldsymbol{\theta}}(\mathbf{x}|\mathbf{z})$ is parametrized with a strong autoregressive neural network, for example, an LSTM (Hochreiter & Schmidhuber, 1997) on text or a PixelCNN (van den Oord et al., 2016) on images. Posterior collapse is especially evident when modeling discrete data, which hinders the usage of VAEs in

---

[1]Code and data are available at https://github.com/jxhe/vae-lagging-encoder

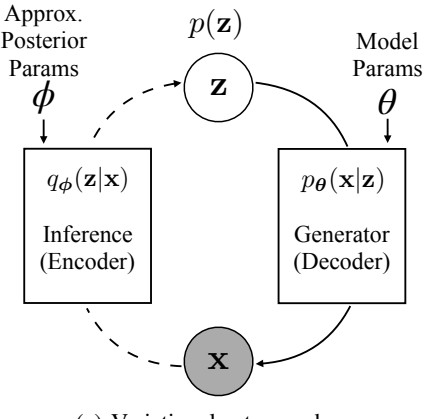

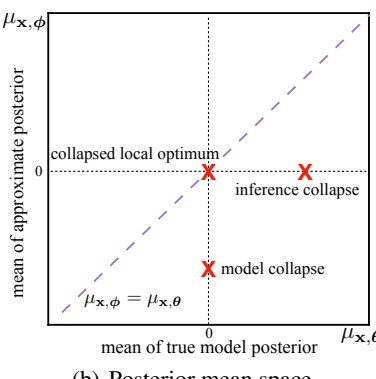

(a) Variational autoencoders

(b) Posterior mean space

Figure 1: **Left:** Depiction of generative model $p(\mathbf{z})p_{\boldsymbol{\theta}}(\mathbf{x}|\mathbf{z})$ and inference network $q_{\boldsymbol{\phi}}(\mathbf{z}|\mathbf{x})$ in VAEs. **Right:** A toy posterior mean space $(\mu_{\mathbf{x},\boldsymbol{\theta}}, \mu_{\mathbf{x},\boldsymbol{\phi}})$ with scalar $z$. The horizontal axis represents the mean of the model posterior $p_{\boldsymbol{\theta}}(\mathbf{z}|\mathbf{x})$, and the vertical axis represents the mean of the approximate posterior $q_{\boldsymbol{\phi}}(\mathbf{z}|\mathbf{x})$. The dashed diagonal line represents when the approximate posterior matches the true model posterior in terms of mean.

important applications like natural language processing. Existing work analyzes this problem from a static optimization perspective, noting that the collapsed solution is often a reasonably good local optimum in terms of ELBO (Chen et al., 2017; Zhao et al., 2017; Alemi et al., 2018). Thus, many proposed solutions to posterior collapse focus on weakening the generator by replacing it with a non-recurrent alternative (Yang et al., 2017; Semeniuta et al., 2017) or modifying the training objective (Zhao et al., 2017; Tolstikhin et al., 2018). In this paper, we analyze the problem from the perspective of training dynamics and propose a novel training procedure for VAEs that addresses posterior collapse. In contrast with other solutions, our proposed procedure optimizes the standard ELBO objective and does not require modification to the VAE model or its parameterization.

Recently, Kim et al. (2018) proposed a new approach to training VAEs by composing the standard inference network with additional mean-field updates. The resulting semi-amortized approach empirically avoided collapse and obtained better ELBO. However, because of the costly instance-specific local inference steps, the new method is more than 10x slower than basic VAE training in practice. It is also unclear why the basic VAE method fails to find better local optima that make use of latents. We consider two questions in this paper: (1) Why does basic VAE training often fall into undesirable collapsed local optima? (2) Is there a simpler way to change the training trajectory to find a non-trivial local optimum?

To this end, we first study the posterior collapse problem from the perspective of training dynamics. We find, empirically, that the posterior approximation often lags far behind the true model posterior in the initial stages of training (Section 3). We then demonstrate how such lagging behavior can drive the generative model towards a collapsed local optimum, and propose a novel training procedure for VAEs that aggressively optimizes the inference network with more updates to mitigate lag (Section 4). Without introducing new modeling components over basic VAEs or additional complexity, our approach is surprisingly simple yet effective in circumventing posterior collapse. As a density estimator, it outperforms neural autoregressive baselines on both text (Yahoo and Yelp) and image (OMNIGLOT) benchmarks, leading to comparable performance with more complicated previous state-of-the-art methods at a fraction of the training cost (Section 6).

## 2 BACKGROUND

### 2.1 VARIATIONAL AUTOENCODERS

VAEs learn deep generative models defined by a prior $p(\mathbf{z})$ and a conditional distribution $p_{\boldsymbol{\theta}}(\mathbf{x}|\mathbf{z})$ as shown in Figure 1(a). In most cases the marginal data likelihood is intractable, so VAEs instead

optimize a tractable variational lower bound (ELBO) of $\log p_{\boldsymbol{\theta}}(\mathbf{x})$,

$$\mathcal{L}(\mathbf{x}; \boldsymbol{\theta}, \boldsymbol{\phi}) = \underbrace{\mathbb{E}_{\mathbf{z} \sim q_{\boldsymbol{\phi}}(\mathbf{z}|\mathbf{x})}[\log p_{\boldsymbol{\theta}}(\mathbf{x}|\mathbf{z})]}_{\text{Reconstruction Loss}} - \underbrace{D_{\text{KL}}(q_{\boldsymbol{\phi}}(\mathbf{z}|\mathbf{x}) \| p(\mathbf{z}))}_{\text{KL Regularizer}}, \tag{2}$$

where $q_{\boldsymbol{\phi}}(\mathbf{z}|\mathbf{x})$ is a variational distribution parameterized by an inference network with parameters $\boldsymbol{\phi}$, and $p_{\boldsymbol{\theta}}(\mathbf{x}|\mathbf{z})$ denotes the generator network with parameters $\boldsymbol{\theta}$. $q_{\boldsymbol{\phi}}(\mathbf{z}|\mathbf{x})$ is optimized to approximate the model posterior $p_{\boldsymbol{\theta}}(\mathbf{z}|\mathbf{x})$. This lower bound is composed of a reconstruction loss term that encourages the inference network to encode information necessary to generate the data and a KL regularizer to push $q_{\boldsymbol{\phi}}(\mathbf{z}|\mathbf{x})$ towards the prior $p(\mathbf{z})$. Below, we consider $p(\mathbf{z}) \coloneqq \mathcal{N}(\mathbf{0}, \boldsymbol{I})$ unless otherwise specified. A key advantage of using inference networks (also called amortized inference) to train deep generative models over traditional locally stochastic variational inference (Hoffman et al., 2013) is that they share parameters over all data samples, amortizing computational cost and allowing for efficient training.

The term VAE is often used both to denote the class of generative models and the amortized inference procedure used in training. In this paper, it is important to distinguish the two and throughout we will refer to the generative model as the *VAE model*, and the training procedure as *VAE training*.

## 2.2 POSTERIOR COLLAPSE

Despite VAE's appeal as a tool to learn unsupervised representations through the use of latent variables, as mentioned in the introduction, VAE models are often found to ignore latent variables when using flexible generators like LSTMs (Bowman et al., 2016). This problem of "posterior collapse" occurs when the training procedure falls into the trivial local optimum of the ELBO objective in which both the variational posterior and true model posterior collapse to the prior. This is undesirable because an important goal of VAEs is to learn meaningful latent features for inputs. Mathematically, posterior collapse represents a local optimum of VAEs where $q_{\boldsymbol{\phi}}(\mathbf{z}|\mathbf{x}) = p_{\boldsymbol{\theta}}(\mathbf{z}|\mathbf{x}) = p(\mathbf{z})$ for all $\mathbf{x}$. To facilitate our analysis about the causes leading up to collapse, we further define two partial collapse states: *model collapse*, when $p_{\boldsymbol{\theta}}(\mathbf{z}|\mathbf{x}) = p(\mathbf{z})$, and *inference collapse*, when $q_{\boldsymbol{\phi}}(\mathbf{z}|\mathbf{x}) = p(\mathbf{z})$ for all $\mathbf{x}$. Note that in this paper we use these two terms to denote the posterior states in the middle of training instead of local optima at the end. These two partial collapse states may not necessarily happen at the same time, which we will discuss later.

## 2.3 VISUALIZATION OF POSTERIOR DISTRIBUTION

Posterior collapse is closely related to the true model posterior $p_{\boldsymbol{\theta}}(\mathbf{z}|\mathbf{x})$ and the approximate posterior $q_{\boldsymbol{\phi}}(\mathbf{z}|\mathbf{x})$ as it is defined. Thus, in order to observe how posterior collapse happens, we track the state of $p_{\boldsymbol{\theta}}(\mathbf{z}|\mathbf{x})$ and $q_{\boldsymbol{\phi}}(\mathbf{z}|\mathbf{x})$ over the course of training, and analyze the training trajectory in terms of the *posterior mean space* $\mathcal{U} = \{\mu : \mu = (\mu_{\mathbf{x},\boldsymbol{\theta}}^T, \mu_{\mathbf{x},\boldsymbol{\phi}}^T)\}$, where $\mu_{\mathbf{x},\boldsymbol{\theta}}$ and $\mu_{\mathbf{x},\boldsymbol{\phi}}$ are the means of $p_{\boldsymbol{\theta}}(\mathbf{z}|\mathbf{x})$ and $q_{\boldsymbol{\phi}}(\mathbf{z}|\mathbf{x})$, respectively.[2] We can then roughly consider $\mu_{\mathbf{x},\boldsymbol{\theta}} = \mathbf{0}$ as model collapse and $\mu_{\mathbf{x},\boldsymbol{\phi}} = \mathbf{0}$ as inference collapse as we defined before. Each $\mathbf{x}$ will be projected to a point in this space under the current model and inference network parameters. If $\mathbf{z}$ is a scalar we can efficiently compute $\mu_{\mathbf{x},\boldsymbol{\theta}}$ and visualize the posterior mean space as shown in Figure 1(b). The diagonal line $\mu_{\mathbf{x},\boldsymbol{\theta}} = \mu_{\mathbf{x},\boldsymbol{\phi}}$ represents parameter settings where $q_{\boldsymbol{\phi}}(\mathbf{z}|\mathbf{x})$ is equal to $p_{\boldsymbol{\theta}}(\mathbf{z}|\mathbf{x})$ in terms of mean, indicating a well-trained inference network. The collapsed local optimum is located at the origin,[3] while the data points at a more desirable local optima may be distributed along the diagonal. In this paper we will utilize this posterior mean space multiple times to analyze the posterior dynamics.

## 3 A LAGGING INFERENCE NETWORK PREVENTS USING LATENT CODES

In this section we analyze posterior collapse from a perspective of training dynamics. We will answer the question of why the basic VAE training with strong decoders tends to hit a collapsed local optimum and provide intuition for the simple solution we propose in Section 4.

---

[2]$\mu_{\mathbf{x},\boldsymbol{\theta}}$ can be approximated through discretization of the model posterior, which we show in Appendix A.

[3]Note that the converse is not true: the setting where all points are located at the origin may not be a local optimum. For example when a model is initialized at the origin as we show in Section 3.2.

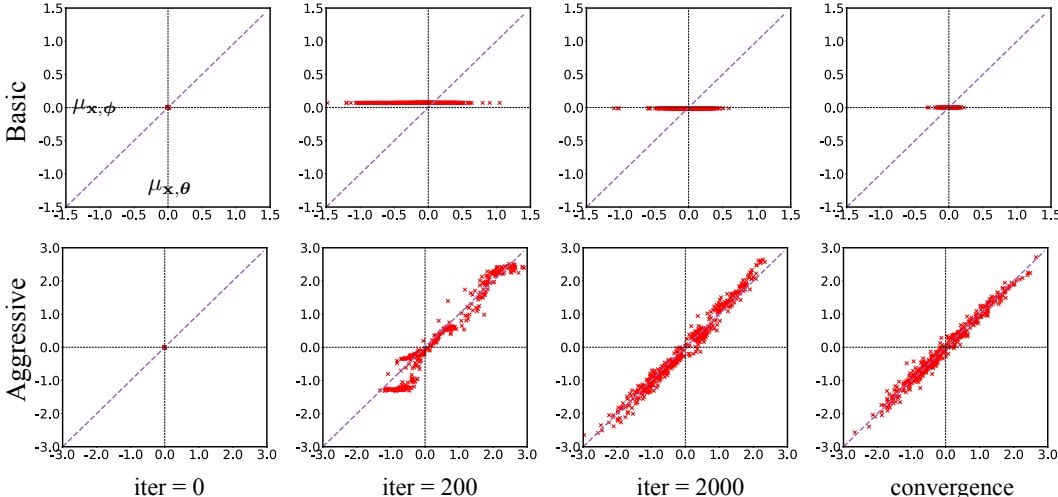

Figure 2: The projections of 500 data samples from a synthetic dataset on the posterior mean space over the course of training. "iter" denotes the number of updates of generators. The top row is from the basic VAE training, the bottom row is from our aggressive inference network training. The results show that while the approximate posterior is lagging far behind the true model posterior in basic VAE training, our aggressive training approach successfully moves the points onto the diagonal line and away from inference collapse.

## 3.1 INTUITIONS FROM ELBO

Since posterior collapse is directly relevant to the approximate posterior $q_\phi(\mathbf{z}|\mathbf{x})$ and true model posterior $p_\theta(\mathbf{z}|\mathbf{x})$, we aim to analyze their training dynamics to study how posterior collapse happens. To this end, it is useful to analyze an alternate form of ELBO:

$$\mathcal{L}(\mathbf{x}; \boldsymbol{\theta}, \boldsymbol{\phi}) = \underbrace{\log p_\theta(\mathbf{x})}_{\text{marginal log data likelihood}} - \underbrace{D_{\mathrm{KL}}(q_\phi(\mathbf{z}|\mathbf{x})\|p_\theta(\mathbf{z}|\mathbf{x}))}_{\text{agreement between approximate and model posteriors}}, \tag{3}$$

With this view, the only goal of approximate posterior $q_\phi(\mathbf{z}|\mathbf{x})$ is to match model posterior $p_\theta(\mathbf{z}|\mathbf{x})$, while the optimization of $p_\theta(\mathbf{z}|\mathbf{x})$ is influenced by two forces, one of which is the ideal objective marginal data likelihood, and the other is $D_{\mathrm{KL}}(q_\phi(\mathbf{z}|\mathbf{x})\|p_\theta(\mathbf{z}|\mathbf{x}))$, which drives $p_\theta(\mathbf{z}|\mathbf{x})$ towards $q_\phi(\mathbf{z}|\mathbf{x})$. Ideally if the approximate posterior is perfect, the second force will vanish, with $\nabla_\theta D_{\mathrm{KL}}(q_\phi(\mathbf{z}|\mathbf{x})|p_\theta(\mathbf{z}|\mathbf{x})) = 0$ when $q_\phi(\mathbf{z}|\mathbf{x}) = p_\theta(\mathbf{z}|\mathbf{x})$. At the start of training, $\mathbf{z}$ and $\mathbf{x}$ are nearly independent under both $q_\phi(\mathbf{z}|\mathbf{x})$ and $p_\theta(\mathbf{z}|\mathbf{x})$ as we show in Section 3.2, i.e. all $\mathbf{x}$ suffer from model collapse in the beginning. Then the only component in the training objective that possibly causes dependence between $\mathbf{z}$ and $\mathbf{x}$ under $p_\theta(\mathbf{z}|\mathbf{x})$ is $\log p_\theta(\mathbf{x})$. However, this pressure may be overwhelmed by the KL term when $p_\theta(\mathbf{z}|\mathbf{x})$ and $q_\phi(\mathbf{z}|\mathbf{x})$ start to diverge but $\mathbf{z}$ and $\mathbf{x}$ remain independent under $q_\phi(\mathbf{z}|\mathbf{x})$. We hypothesize that, in practice, training drives $p_\theta(\mathbf{z}|\mathbf{x})$ and $q_\phi(\mathbf{z}|\mathbf{x})$ to the prior in order to bring them into alignment, while locking into model parameters that capture the distribution of $\mathbf{x}$ while ignoring $\mathbf{z}$. Critically, posterior collapse is a *local optimum*; once a set of parameters that achieves these goals are reached, gradient optimization fails to make further progress, even if better overall models that make use of $\mathbf{z}$ to describe $\mathbf{x}$ exist.

Next we visualize the posterior mean space by training a basic VAE with a scalar latent variable on a relatively simple synthetic dataset to examine our hypothesis.

## 3.2 OBSERVATIONS ON SYNTHETIC DATA

As a synthetic dataset we use discrete sequence data since posterior collapse has been found the most severe in text modeling tasks. Details on this synthetic dataset and experiment are in Appendix B.1.

We train a basic VAE with a scalar latent variable, LSTM encoder, and LSTM decoder on our synthetic dataset. We sample 500 data points from the validation set and show them on the posterior mean space plots at four different training stages from initialization to convergence in Figure 2. The mean of the approximate posterior distribution $\mu_{\mathbf{x},\phi}$ is from the output of the inference network, and $\mu_{\mathbf{x},\theta}$ can be approximated by discretization of the true model posterior $p_\theta(\mathbf{z}|\mathbf{x})$ (see Appendix A).

**Algorithm 1** VAE training with controlled aggressive inference network optimization.

1: $\boldsymbol{\theta}, \boldsymbol{\phi} \leftarrow$ Initialize parameters
2: $aggressive \leftarrow$ TRUE
3: **repeat**
4:     **if** $aggressive$ **then**
5:         **repeat**             ▷ [aggressive updates]
6:             $\mathbf{X} \leftarrow$ Random data minibatch
7:             Compute gradients $\boldsymbol{g}_{\boldsymbol{\phi}} \leftarrow \nabla_{\boldsymbol{\phi}} \mathcal{L}(\mathbf{X}; \boldsymbol{\theta}, \boldsymbol{\phi})$
8:             Update $\boldsymbol{\phi}$ using gradients $\boldsymbol{g}_{\boldsymbol{\phi}}$
9:         **until** convergence
10:         $\mathbf{X} \leftarrow$ Random data minibatch
11:         Compute gradients $\boldsymbol{g}_{\boldsymbol{\theta}} \leftarrow \nabla_{\boldsymbol{\theta}} \mathcal{L}(\mathbf{X}; \boldsymbol{\theta}, \boldsymbol{\phi})$
12:         Update $\boldsymbol{\theta}$ using gradients $\boldsymbol{g}_{\boldsymbol{\theta}}$
13:     **else**             ▷ [basic VAE training]
14:         $\mathbf{X} \leftarrow$ Random data minibatch
15:         Compute gradients $\boldsymbol{g}_{\boldsymbol{\theta}, \boldsymbol{\phi}} \leftarrow \nabla_{\boldsymbol{\phi}, \boldsymbol{\theta}} \mathcal{L}(\mathbf{X}; \boldsymbol{\theta}, \boldsymbol{\phi})$
16:         Update $\boldsymbol{\theta}, \boldsymbol{\phi}$ using $\boldsymbol{g}_{\boldsymbol{\theta}, \boldsymbol{\phi}}$
17:     **end if**
18:     Update $aggressive$ as discussed in Section 4.2
19: **until** convergence

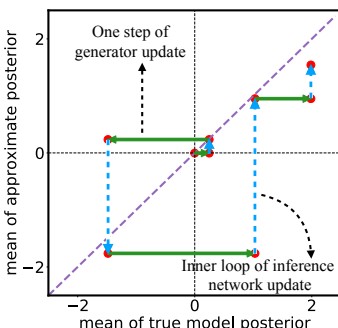

Figure 3: Trajectory of one data instance on the posterior mean space with our aggressive training procedure. Horizontal arrow denotes one step of generator update, and vertical arrow denotes the inner loop of inference network update. We note that the approximate posterior $q_{\boldsymbol{\phi}}(\mathbf{z}|\mathbf{x})$ takes an aggressive step to catch up to the model posterior $p_{\boldsymbol{\theta}}(\mathbf{z}|\mathbf{x})$.

As illustrated in Figure 2, all points are located at the origin upon initialization[4], which means $\mathbf{z}$ and $\mathbf{x}$ are almost independent in terms of both $q_{\boldsymbol{\phi}}(\mathbf{z}|\mathbf{x})$ and $p_{\boldsymbol{\theta}}(\mathbf{z}|\mathbf{x})$ at the beginning of training. In the second stage of basic VAE training, the points start to spread along the $\mu_{\mathbf{x}, \boldsymbol{\theta}}$ axis. This phenomenon implies that for some data points $p_{\boldsymbol{\theta}}(\mathbf{z}|\mathbf{x})$ moves far away from the prior $p(\mathbf{z})$, and confirms that $\log p_{\boldsymbol{\theta}}(\mathbf{x})$ is able to help move away from model collapse. However, all of these points are still distributed along a horizontal line, which suggests that $q_{\boldsymbol{\phi}}(\mathbf{z}|\mathbf{x})$ fails to catch up to $p_{\boldsymbol{\theta}}(\mathbf{z}|\mathbf{x})$ and these points are still in a state of inference collapse. As expected, the dependence between $\mathbf{z}$ and $\mathbf{x}$ under $p_{\boldsymbol{\theta}}(\mathbf{z}|\mathbf{x})$ is gradually lost and finally the model converges to the collapsed local optimum.

## 4 METHOD

### 4.1 AGGRESSIVE TRAINING OF THE INFERENCE NETWORK

The problem reflected in Figure 2 implies that the inference network is lagging far behind $p_{\boldsymbol{\theta}}(\mathbf{z}|\mathbf{x})$, and might suggest more "aggressive" inference network updates are needed. Instead of blaming the poor approximation on the limitation of the inference network's amortization, we hypothesize that the optimization of the inference and generation networks are imbalanced, and propose to separate the optimization of the two. Specifically, we change the training procedure to:

$$\boldsymbol{\theta}^* = \arg\max_{\boldsymbol{\theta}} \ \mathcal{L}(\mathbf{X}; \boldsymbol{\theta}, \boldsymbol{\phi}^*), \text{ where } \boldsymbol{\phi}^* = \arg\max_{\boldsymbol{\phi}} \ \mathcal{L}(\mathbf{X}; \boldsymbol{\theta}, \boldsymbol{\phi}), \tag{4}$$

where optimizing the inference network $q_{\boldsymbol{\phi}}(\mathbf{z}|\mathbf{x})$ is an inner loop in the entire training process as shown in Algorithm 1. This training procedure shares the same spirit with traditional stochastic variational inference (SVI) (Hoffman et al., 2013) that performs iterative inference for each data point separately and suffers from very lengthy iterative estimation. Compared with recent work that try to combine amortized variational inference and SVI (Hjelm et al., 2016; Krishnan et al., 2018; Kim et al., 2018; Marino et al., 2018) where the inference network is learned to be a component to help perform instance-specific variational inference, our approach keeps variational inference fully amortized, allowing for reverting back to efficient basic VAE training as discussed in Section 4.2. Also, this aggressive inference network optimization algorithm is as simple as basic VAE training without introducing additional SVI steps, yet attains comparable performance to more sophisticated approaches as we will show in Section 6.

---

[4]In Appendix G we also study the setting where the points are not initialized at origin.

## 4.2 STOPPING CRITERION

Always training with Eq.4 would be inefficient and neglects the benefit of the amortized inference network. Following our previous analysis, the term $D_{\mathrm{KL}}(q_\phi(\mathbf{z}|\mathbf{x})\|p_\theta(\mathbf{z}|\mathbf{x}))$ tends to pressure $q_\phi(\mathbf{z}|\mathbf{x})$ or $p_\theta(\mathbf{z}|\mathbf{x})$ to $p(\mathbf{z})$ only if at least one of them is close to $p(\mathbf{z})$, and thus we posit that if we can confirm that we haven't reached this degenerate condition, we can continue with standard VAE training. Since $q_\phi(\mathbf{z}|\mathbf{x})$ is the one lagging behind, we use the mutual information $I_q$ between $\mathbf{z}$ and $\mathbf{x}$ under $q_\phi(\mathbf{z}|\mathbf{x})$ to control our stopping criterion. In practice, we compute the mutual information on the validation set every epoch, and stop the aggressive updates when $I_q$ stops climbing. In all our experiments in this paper we found that the aggressive algorithm usually reverts back to basic VAE training within 5 epochs. Mutual information, $I_q$ can be computed by (Hoffman & Johnson, 2016):

$$I_q = \mathbb{E}_{\mathbf{x} \sim p_d(\mathbf{x})}[D_{\mathrm{KL}}(q_\phi(\mathbf{z}|\mathbf{x})\|p(\mathbf{z}))] - D_{\mathrm{KL}}(q_\phi(\mathbf{z})\|p(\mathbf{z})), \tag{5}$$

where $p_d(\mathbf{x})$ is the empirical distribution. The aggregated posterior, $q_\phi(\mathbf{z}) = \mathbb{E}_{\mathbf{x} \sim p_d(\mathbf{x})}[q_\phi(\mathbf{z}|\mathbf{x})]$, can be approximated with a Monte Carlo estimate. $D_{\mathrm{KL}}(q_\phi(\mathbf{z})\|p(\mathbf{z}))$ is also approximated by Monte Carlo, where samples from $q_\phi(\mathbf{z})$ can be easily obtained by ancestral sampling (i.e. sample $\mathbf{x}$ from dataset and sample $\mathbf{z} \sim q_\phi(\mathbf{z}|\mathbf{x})$). This estimator for $I_q$ is the same as in (Dieng et al., 2018), which is biased because the estimation for $D_{\mathrm{KL}}(q_\phi(\mathbf{z})\|p(\mathbf{z}))$ is biased. More specifically, it is a Monte Carlo estimate of an upper bound of mutual information. The complete algorithm is shown in Algorithm 1.

## 4.3 OBSERVATIONS ON SYNTHETIC DATASET

By training the VAE model with our approach on synthetic data, we visualize the 500 data samples in the posterior mean space in Figure 2. From this, it is evident that the points move towards $\mu_{\mathbf{x},\theta} = \mu_{\mathbf{x},\phi}$ and are roughly distributed along the diagonal in the end. This is in striking contrast to the basic VAE and confirms our hypothesis that the inference and generator optimization can be rebalanced by simply performing more updates of the inference network. In Figure 3 we show the training trajectory of one single data instance for the first several optimization iterations and observe how the aggressive updates help escape inference collapse.

## 5 RELATION TO RELATED WORK

Posterior collapse in VAEs is first detailed in (Bowman et al., 2016) where they combine a LSTM decoder with VAE for text modeling. They interpret this problem from a regularization perspective, and propose the "KL cost annealing" method to address this issue, whereby the weight of KL term between approximate posterior and prior increases from a small value to one in a "warm-up" period. This method has been shown to be unable to deal with collapse on complex text datasets with very large LSTM decoders (Yang et al., 2017; Kim et al., 2018). Many works follow this line to lessen the effect of KL term such as $\beta$-VAE (Higgins et al., 2017) that treats the KL weight as a hyperparameter or "free bits" method that constrains the minimum value of the KL term. Our approach differs from these methods in that we do not change ELBO objective during training and are in principle still performing maximum likelihood estimation. While these methods explicitly encourage the use of the latent variable, they may implicitly sacrifice density estimation performance at the same time, as we will discuss in Section 6.

Another thread of research focuses on a different problem called the "amortization gap" (Cremer et al., 2018), which refers to the difference of ELBO caused by parameter sharing of the inference network. Some approaches try to combine instance-specific variational inference with amortized variational inference to narrow this gap (Hjelm et al., 2016; Krishnan et al., 2018; Kim et al., 2018; Marino et al., 2018). The most related example is SA-VAE (Kim et al., 2018), which mixes instance-specific variational inference and empirically avoids posterior collapse. Our approach is much simpler without sacrificing performance, yet achieves an average of 5x training speedup.

Other attempts to address posterior collapse include proposing new regularizers (Zhao et al., 2017; Tolstikhin et al., 2018; Phuong et al., 2018), deploying less powerful decoders (Yang et al., 2017; Semeniuta et al., 2017), using lossy input (Chen et al., 2017), utilizing different latent variable connections (Dieng et al., 2017; 2018; Park et al., 2018), or changing the prior (Tomczak & Welling, 2018; Xu & Durrett, 2018).

Table 1: Results on Yahoo and Yelp datasets. We report mean values across 5 different random restarts, and standard deviation is given in parentheses when available. For LSTM-LM* we report the exact negative log likelihood.

| Model | Yahoo | | | | Yelp | | | |
|---|---|---|---|---|---|---|---|---|
| | NLL | KL | MI | AU | NLL | KL | MI | AU |
| **Previous Reports** | | | | | | | | |
| CNN-VAE (Yang et al., 2017) | ≤332.1 | 10.0 | – | – | ≤359.1 | 7.6 | – | – |
| SA-VAE + anneal (Kim et al., 2018) | ≤327.5 | 7.19 | – | – | – | – | – | – |
| **Modified VAE Objective** | | | | | | | | |
| VAE + anneal | 328.6 (0.0) | 0.0 (0.0) | 0.0 (0.0) | 0.0 (0.0) | 357.9 (0.1) | 0.0 (0.0) | 0.0 (0.0) | 0.0 (0.0) |
| $\beta$-VAE ($\beta = 0.2$) | 332.2 (0.6) | 19.1 (1.5) | 3.3 (0.1) | 20.4 (6.8) | 360.7 (0.7) | 11.7 (2.4) | 3.0 (0.5) | 10.0 (5.9) |
| $\beta$-VAE ($\beta = 0.4$) | 328.7 (0.1) | 6.3 (1.7) | 2.8 (0.6) | 8.0 (5.2) | 358.2 (0.3) | 4.2 (0.4) | 2.0 (0.3) | 4.2 (3.8) |
| $\beta$-VAE ($\beta = 0.6$) | 328.5 (0.1) | 0.3 (0.2) | 0.2 (0.1) | 1.0 (0.7) | 357.9 (0.1) | 0.2 (0.2) | 0.1 (0.1) | 3.8 (2.9) |
| $\beta$-VAE ($\beta = 0.8$) | 328.8 (0.1) | 0.0 (0.0) | 0.0 (0.0) | 0.0 (0.0) | 358.1 (0.2) | 0.0 (0.0) | 0.0 (0.0) | 0.0 (0.0) |
| SA-VAE + anneal | 327.2 (0.2) | 5.2 (1.4) | 2.7 (0.5) | 9.8 (1.3) | 355.9 (0.1) | 2.8 (0.5) | 1.7 (0.3) | 8.4 (0.9) |
| Ours + anneal | **326.7 (0.1)** | 5.7 (0.7) | 2.9 (0.2) | 15.0 (3.5) | **355.9 (0.1)** | 3.8 (0.2) | 2.4 (0.1) | 11.3 (1.0) |
| **Standard VAE Objective** | | | | | | | | |
| LSTM-LM* | **328.0 (0.3)** | – | – | – | 358.1 (0.6) | – | – | – |
| VAE | 329.0 (0.1) | 0.0 (0.0) | 0.0 (0.0) | 0.0 (0.0) | 358.3 (0.2) | 0.0 (0.0) | 0.0 (0.0) | 0.0 (0.0) |
| SA-VAE | 329.2 (0.2) | 0.1 (0.0) | 0.1 (0.0) | 0.8 (0.4) | 357.8 (0.2) | 0.3 (0.1) | 0.3 (0.0) | 1.0 (0.0) |
| Ours | 328.2 (0.2) | 5.6 (0.2) | 3.0 (0.0) | 8.0 (0.0) | **356.9 (0.2)** | 3.4 (0.3) | 2.4 (0.1) | 7.4 (1.3) |

Table 2: Results on OMNIGLOT dataset. We report mean values across 5 different random restarts, and standard deviation is given in parentheses when available. For PixelCNN* we report the exact negative log likelihood.

| Model | NLL | KL | MI | AU |
|---|---|---|---|---|
| **Previous Reports** | | | | |
| VLAE (Chen et al., 2017) | 89.83 | – | – | – |
| VampPrior (Tomczak & Welling, 2018) | 89.76 | – | – | – |
| **Modified VAE Objective** | | | | |
| VAE + anneal | 89.21 (0.04) | 1.97 (0.12) | 1.79 (0.11) | 5.3 (1.0) |
| $\beta$-VAE ($\beta = 0.2$) | 105.96 (0.38) | 69.62 (2.16) | 3.89 (0.03) | 32.0 (0.0) |
| $\beta$-VAE ($\beta = 0.4$) | 96.09 (0.36) | 44.93 (12.17) | 3.91 (0.03) | 32.0 (0.0) |
| $\beta$-VAE ($\beta = 0.6$) | 92.14 (0.12) | 25.43 (9.12) | 3.93 (0.03) | 32.0 (0.0) |
| $\beta$-VAE ($\beta = 0.8$) | 89.15 (0.04) | 9.98 (0.20) | 3.84 (0.03) | 13.0 (0.7) |
| SA-VAE + anneal | **89.07 (0.06)** | 3.32 (0.08) | 2.63 (0.04) | 8.6 (0.5) |
| Ours + anneal | 89.11 (0.04) | 2.36 (0.15) | 2.02 (0.12) | 7.2 (1.3) |
| **Standard VAE Objective** | | | | |
| PixelCNN* | 89.73 (0.04) | – | – | – |
| VAE | 89.41 (0.04) | 1.51 (0.05) | 1.43 (0.07) | 3.0 (0.0) |
| SA-VAE | 89.29 (0.02) | 2.55 (0.05) | 2.20 (0.03) | 4.0 (0.0) |
| Ours | **89.05 (0.05)** | 2.51 (0.14) | 2.19 (0.08) | 5.6 (0.5) |

# 6 EXPERIMENTS

Our experiments below are designed to (1) examine whether the proposed method indeed prevents posterior collapse, (2) test its efficacy with respect to maximizing predictive log-likelihood compared to other existing approaches, and (3) test its training efficiency.

## 6.1 SETUP

For all experiments we use a Gaussian prior $\mathcal{N}(\mathbf{0}, \mathbf{I})$ and the inference network parametrizes a diagonal Gaussian. We evaluate with approximate negative log likelihood (NLL) as estimated by 500 importance weighted samples[5] (Burda et al., 2016) since it produces a tighter lower bound to marginal data log likelihood than ELBO (ELBO values are included in Appendix C), and should be more accurate. We also report $D_{\mathrm{KL}}(q_\phi(\mathbf{z}|\mathbf{x})\|p(\mathbf{z}))$ (KL), mutual information $I_q$ (MI), and number of active units (AU) (Burda et al., 2016) in latent representation. The activity of a latent dimension $z$ is measured as $A_z = \mathrm{Cov}_{\mathbf{x}}(\mathbb{E}_{z \sim q(z|\mathbf{x})}[z])$. The dimension $z$ is defined as active if $A_z > 0.01$.

---

[5]We measure the uncertainty in the evaluation caused by the Monte Carlo estimates in Appendix D. The variance of our NLL estimates for a trained VAE model is smaller than $10^{-3}$ on all datasets.

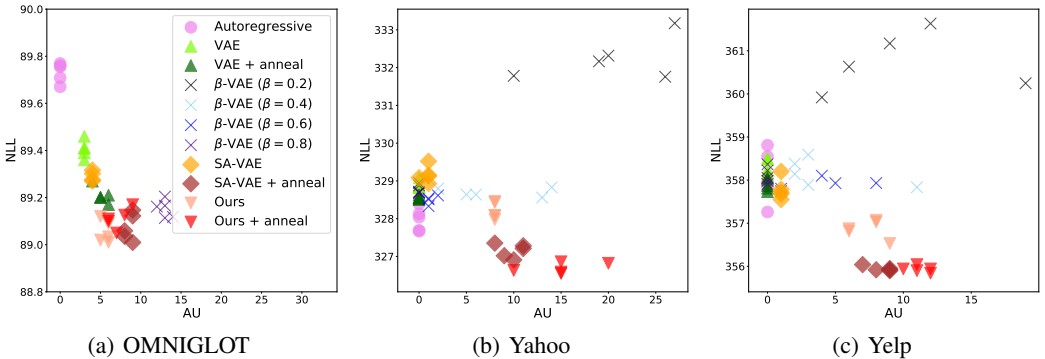

(a) OMNIGLOT  (b) Yahoo  (c) Yelp

Figure 4: NLL versus AU (active units) for all models on three datasets. For each model we display 5 points which represent 5 runs with different random seeds. "Autoregressive" denotes LSTM-LM for text data and PixelCNN for image data. We plot "autoregressive" baselines as their AU is 0. To better visualize the system difference on OMNIGLOT dataset, for OMNIGLOT figure we ignore some $\beta$-VAE baselines that are not competitive.

As baselines, we compare with strong neural autoregressive models (LSTM-LM for text and Pixel-CNN (van den Oord et al., 2016) for images), basic VAE, the "KL cost annealing" method (Bowman et al., 2016; Sønderby et al., 2016), $\beta$-VAE (Higgins et al., 2017), and SA-VAE (Kim et al., 2018) which holds the previous state-of-the-art performance on text modeling benchmarks. For $\beta$-VAE we vary $\beta$ between 0.2, 0.4, 0.6, and 0.8. SA-VAE is ran with 10 refinement steps. We also examine the effect of KL cost annealing on both SA-VAE and our approach. To facilitate our analysis later, we report the results in two categories: "Standard VAE objectives", and "Modified VAE objectives".[6]

We evaluate our method on density estimation for text on the Yahoo and Yelp corpora (Yang et al., 2017) and images on OMNIGLOT (Lake et al., 2015). Following the same configuration as in Kim et al. (2018), we use a single layer LSTM as encoder and decoder for text. For images, we use a ResNet (He et al., 2016) encoder and a 13-layer Gated PixelCNN (van den Oord et al., 2016) decoder. We use 32-dimensional $z$ and optimize ELBO objective with SGD for text and Adam (Kingma & Ba, 2015) for images. We concatenate $z$ to the input for the decoders. For text, $z$ also predicts the initial hidden state of the LSTM decoder. We dynamically binarize images during training and test on fixed binarized test data. We run all models with 5 different random restarts, and report mean and standard deviation. Full details of the setup are in Appendix B.2 and B.3.

## 6.2 RESULTS

In Table 1 and Table 2 we show the results on all three datasets, we also plot NLL vs AU for every trained model from separate runs in Figure 4 to visualize the uncertainties. Our method achieves comparable or better performance than previous state-of-the-art systems on all three datasets. Note that to examine the posterior collapse issue for images we use a larger PixelCNN decoder than previous work, thus our approach is not directly comparable to them and included at the top of Table 2 as reference points. We observe that SA-VAE suffers from posterior collapse on both text datasets without annealing. However, we demonstrate that our algorithm does not experience posterior collapse even without annealing.

## 6.3 TRAINING TIME

In Table 3 we report the total training time of our approach, SA-VAE and basic VAE training across the three datasets. We find that the training time for our algorithm is only 2-3 times slower than a regular VAE whilst being 3-7 times faster than SA-VAE.

---

[6]While annealing reverts back to ELBO objective after the warm-up period, we consider part of "Modified VAE objectives" since it might produce undesired behavior in the warm-up period, as we will discuss soon.

Table 3: Comparison of total training time, in terms of relative speed and absolute hours.

| | Yahoo | | Yelp15 | | OMNIGLOT | |
|---|---|---|---|---|---|---|
| | **Relative** | **Hours** | **Relative** | **Hours** | **Relative** | **Hours** |
| VAE | 1.00 | 5.35 | 1.00 | 5.75 | 1.00 | 4.30 |
| SA-VAE | 9.91 | 52.99 | 10.33 | 59.37 | 15.15 | 65.07 |
| Ours | 2.20 | 11.76 | 3.73 | 21.44 | 2.19 | 9.42 |

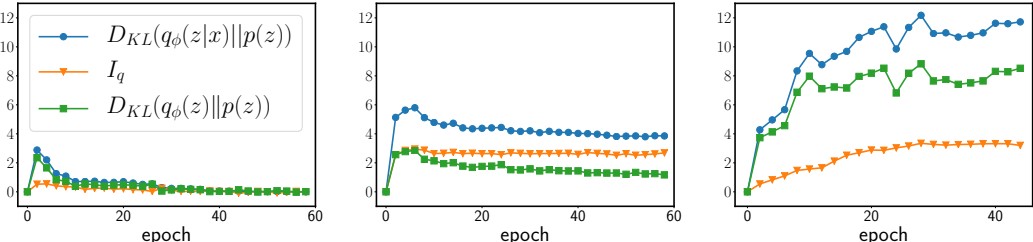

Figure 5: Training behavior on Yelp. **Left:** VAE + annealing. **Middle:** Our method. **Right:** $\beta$-VAE ($\beta = 0.2$).

Table 4: Results on Yelp dataset using a fixed budget of inner encoder updates

| # Inner Iterations | NLL | KL | MI | AU | Hours |
|---|---|---|---|---|---|
| 10 | 357.9 | 1.1 | 1.0 | 3 | 11.97 |
| 30 | 357.1 | 3.6 | 2.5 | 8 | 22.31 |
| 50 | 356.9 | 4.2 | 2.8 | 9 | 29.58 |
| 70 | 357.1 | 4.4 | 2.7 | 10 | 24.18 |
| convergence | 357.0 | 3.8 | 2.6 | 8 | 21.44 |

### 6.4 ANALYSIS OF BASELINES

We analyze the difference between our approach and the methods that weaken the KL regularizer term in ELBO, and explain the unwanted behavior produced by breaking maximum likelihood estimation. As illustrative examples, we compare with the KL cost annealing method and $\beta$-VAE. Decreasing the weight of the KL regularizer term in ELBO is equivalent to adding an additional regularizer to push $q_\phi(\mathbf{z}|\mathbf{x})$ far from $p(\mathbf{z})$. We set $\beta = 0.2$ in order to better observe this phenomenon.

We investigate the training procedure on the Yelp dataset based on: (1) the mutual information between $\mathbf{z}$ and $\mathbf{x}$, $I_q$, (2) the KL regularizer, $\mathbb{E}_{\mathbf{x} \sim p_d(\mathbf{x})}[D_{\mathrm{KL}}(q_\phi(\mathbf{z}|\mathbf{x})\|p(\mathbf{z}))]$, and (3) the distance between the aggregated posterior and the prior, $D_{\mathrm{KL}}(q_\phi(\mathbf{z})\|p(\mathbf{z}))$. Note that the KL regularizer is equal to the sum of the other two as stated in Eq.5. We plot these values over the course of training in Figure 5. In the initial training stage we observe that the KL regularizer increases with all three approaches, however, the mutual information, $I_q$, in the annealing remains small, thus a large KL regularizer term does not imply that the latent variable is being used. Finally the annealing method suffers from posterior collapse. For $\beta$-VAE, the mutual information increases, but $D_{\mathrm{KL}}(q_\phi(\mathbf{z})\|p(\mathbf{z}))$ also reaches a very large value. Intuitively, $D_{\mathrm{KL}}(q_\phi(\mathbf{z})\|p(\mathbf{z}))$ should be kept small for learning the generative model well since in the objective the generator $p_\theta(\mathbf{x}|\mathbf{z})$ is learned with latent variables sampled from the variational distribution. If the setting of $\mathbf{z}$ that best explains the data has a lower likelihood under the model prior, then the overall model would fit the data poorly. The same intuition has been discussed in Zhao et al. (2017) and Tolstikhin et al. (2018). This also explains why $\beta$-VAE generalizes poorly when it has large mutual information. In contrast, our approach is able to obtain high mutual information, and at the same time maintain a small $D_{\mathrm{KL}}(q_\phi(\mathbf{z})\|p(\mathbf{z}))$ as a result of optimizing standard ELBO where the KL regularizer upper-bounds $D_{\mathrm{KL}}(q_\phi(\mathbf{z})\|p(\mathbf{z}))$.

### 6.5 ANALYSIS OF INNER LOOP UPDATE

We perform analysis to examine the tradeoff between performance and speed within the inner loop update in our approach, through fixing a budget of updates to the inference network instead of updat-

ing until convergence.[7] In our implementation, we break the inner loop when the ELBO objective stays the same or decreases across 10 iterations. Note that we do not perform separate learning rate decay in the inner loop so this convergence condition is not strict, but empirically we found it to be sufficient. Across all datasets, in practice this yields roughly 30 – 100 updates per inner loop update. Now we explore using a fixed budget of inner loop updates and observe its influence on performance and speed. We report the results on Yelp dataset from single runs in Table 4.[8] We see that sufficient number of inner iterations is necessary to address posterior collapse and achieve good performance, but the performance starts to saturate near convergence, thus we believe that optimizing to a near-convergence point is important.

# 7 CONCLUSION

In this paper we study the "posterior collapse" problem that variational autoencoders experience when the model is parameterized by a strong autoregressive neural network. In our synthetic experiment we identify that the problem lies with the lagging inference network in the initial stages of training. To remedy this, we propose a simple yet effective training algorithm that aggressively optimizes the inference network with more updates before reverting back to basic VAE training. Experiments on text and image modeling demonstrate the effectiveness of our approach.

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

## A  APPROXIMATION OF THE MEAN OF THE TRUE MODEL POSTERIOR

We approximate the mean of true model posterior $p_{\boldsymbol{\theta}}(\mathbf{z}|\mathbf{x})$ by discretization of the density distribution (Riemann integral):

$$\mathbb{E}_{\mathbf{z} \sim p_{\boldsymbol{\theta}}(\mathbf{z}|\mathbf{x})}[z] = \sum_{z_i \in \mathcal{C}} [z_i p(z_i|\mathbf{x})], \tag{6}$$

where $\mathcal{C}$ is a partition of an interval with small stride and sufficiently large coverage. We assume the density value outside this interval is zero. The model posterior, $p_{\boldsymbol{\theta}}(z|\mathbf{x})$, needs to be first approximated on this partition of interval. In practice, for the synthetic data we choose the interval [-20.0, 20.0] and stride equal to 0.01. This interval should have sufficient coverage since we found all samples from true model posterior $p_{\boldsymbol{\theta}}(z|\mathbf{x})$ lies within [-5.0, 5.0] by performing MH sampling.

## B  EXPERIMENTAL DETAILS

In general, for annealing we increase the KL weight linearly from 0.1 to 1.0 in the first 10 epochs, as in Kim et al. (2018). We also perform analysis for different annealing strategies in Appendix E

### B.1  SYNTHETIC EXPERIMENT FOR SECTION 3 AND 4

To generate synthetic data points, we first sample a two-dimensional latent variable $\mathbf{z}$ from a mixture of Gaussian distributions that have four mixture components. We choose dimension two because we want the synthetic data distribution to be relatively simple but also complex enough for a one-dimensional latent variable model to fit. We choose mixture of Gaussian as the prior to make sure that the synthetic data is diverse. The mean of these Gaussians are (-2.0, -2.0), (-2.0, 2.0), (2.0, -2.0), (2.0, 2.0), respectively. All of them have a unit variance. Then we follow the synthetic data generation procedure in Kim et al. (2018), where we sample data points from an one-layer LSTM conditioned on latent variables. The LSTM has 100 hidden units and 100-dimensional input embeddings. An affine transformation of $\mathbf{z}$ is used as the initial hidden state of LSTM decoder, $\mathbf{z}$ is also concatenated with output of LSTM at each time stamp to be directly mapped to vocabulary space. LSTM parameters are initialized with $\mathcal{U}(-1, 1)$, and the part of MLP that maps $\mathbf{z}$ to vocabulary space is initialized with $\mathcal{U}(-5, 5)$, this is done to make sure that the latent variables have more influence in generating data. We generated a dataset with 20,000 examples (train/val/test is 16000/2000/2000) each of length 10 from a vocabulary of size 1000.

In the synthetic experiment we use a LSTM encoder and LSTM decoder, both of which have 50 hidden units and 50 latent embeddings. This LSTM decoder has less capacity than the one used for creating the dataset since in the real world model capacity is usually insufficient to exactly model the empirical distribution. Parameters of LSTM decoders are initialized with $\mathcal{U}(-0.01, 0.01)$, except for the embedding weight matrix which is initialized with $\mathcal{U}(-0.1, 0.1)$. Dropout layers with probability 0.5 are applied to both input embeddings and output hidden embeddings of decoder. We use the SGD optimizer and start with a learning rate of 1.0 and decay it by a factor of 2 if the validation loss has not improved in 2 epochs and terminate training once the learning rate has decayed a total of 5 times.

### B.2  TEXT

Following Kim et al. (2018), we use a single-layer LSTM with 1024 hidden units and 512-dimensional word embeddings as the encoder and decoder for all of text models. The LSTM parameters are initialized from $\mathcal{U}(-0.01, 0.01)$, and embedding parameters are initialized from $\mathcal{U}(-0.1, 0.1)$. We use the final hidden state of the encoder to predict (via a linear transformation) the latent variable. We use the SGD optimizer and start with a learning rate of 1.0 and decay it by a factor of 2 if the validation loss has not improved in 2 epochs and terminate training once the learning rate has decayed a total of 5 times. We don't perform any text preprocessing and use the datasets as provided. We follow Kim et al. (2018) and use dropout of 0.5 on the decoder for both the input embeddings of the decoder and on the output of the decoder before the linear transformation to vocabulary space.

### B.3 IMAGES

We use the same train/val/test splits as provided by Kim et al. (2018). We use the Adam optimizer and start with a learning rate of 0.001 and decay it by a factor of 2 if the validation loss has not improved in 20 epochs. We terminate training once the learning rate has decayed a total of 5 times. Inputs were dynamically binarized throughout training by viewing the input as Bernoulli random variables that are sampled from pixel values. We validate and test on a fixed binarization and our decoder uses binary likelihood. Our ResNet is the same as used by Chen et al. (2017). Our 13-layer PixelCNN architecture is a larger variant based on what was used in Kim et al. (2018) and described in their Appendix B.3 section. The PixelCNN has five 7 x 7 layers, followed by, four 5 x 5 layers, and then four 3 x 3 layers. Each layer has 64 feature maps. We use batch normalization followed by an ELU activation before our final 1 x 1 convolutional layer and sigmoid nonlinearity.

## C ADDITIONAL RESULTS CONTAINING ELBO

Table 5: Results on Yahoo and Yelp datasets. We report mean values across 5 different random restarts, and standard deviation is given in parentheses when available. For LSTM-LM* we report the exact negative log likelihood.

| Model | Yahoo | | | | | Yelp | | | | |
| --- | --- | --- | --- | --- | --- | --- | --- | --- | --- | --- |
| | IW | -ELBO | KL | MI | AU | IW | -ELBO | KL | MI | AU |
| **Previous Reports** | | | | | | | | | | |
| CNN-VAE (Yang et al., 2017) | – | 332.1 | 10.0 | – | – | – | 359.1 | 7.6 | – | – |
| SA-VAE + anneal (Kim et al., 2018) | – | 327.5 | 7.19 | – | – | – | – | – | – | – |
| **Modified VAE Objective** | | | | | | | | | | |
| VAE + anneal | 328.6 (0.0) | 328.8 (0.0) | 0.0 (0.0) | 0.0 (0.0) | 0.0 (0.0) | 357.9 (0.1) | 358.1 (0.1) | 0.0 (0.0) | 0.0 (0.0) | 0.0 (0.0) |
| $\beta$-VAE ($\beta = 0.2$) | 332.2 (0.6) | 335.9 (0.8) | 19.1 (1.5) | 3.3 (0.1) | 20.4 (6.8) | 360.7 (0.7) | 363.2 (1.1) | 11.7 (2.4) | 3.0 (0.5) | 10.0 (5.9) |
| $\beta$-VAE ($\beta = 0.4$) | 328.7 (0.1) | 330.2 (0.4) | 6.3 (1.7) | 2.8 (0.6) | 8.0 (5.2) | 358.2 (0.3) | 359.1 (0.3) | 4.2 (0.4) | 2.0 (0.3) | 4.2 (3.8) |
| $\beta$-VAE ($\beta = 0.6$) | 328.5 (0.1) | 328.9 (0.0) | 0.3 (0.2) | 0.2 (0.1) | 1.0 (0.7) | 357.9 (0.1) | 358.2 (0.1) | 0.2 (0.2) | 0.1 (0.1) | 3.8 (2.9) |
| $\beta$-VAE ($\beta = 0.8$) | 328.8 (0.1) | 329.0 (0.1) | 0.0 (0.0) | 0.0 (0.0) | 0.0 (0.0) | 358.1 (0.0) | 358.3 (0.2) | 0.0 (0.0) | 0.0 (0.0) | 0.0 (0.0) |
| SA-VAE + anneal | 327.2 (0.2) | 327.8 (0.2) | 5.2 (1.4) | 2.7 (0.5) | 9.8 (1.3) | 355.9 (0.1) | 356.2 (0.1) | 2.8 (0.5) | 1.7 (0.3) | 8.4 (0.9) |
| Ours + anneal | **326.7 (0.1)** | 328.4 (0.2) | 5.7 (0.7) | 2.9 (0.2) | 15.0 (3.5) | **355.9 (0.1)** | 357.2 (0.1) | 3.8 (0.2) | 2.4 (0.1) | 11.3 (1.0) |
| **Standard VAE Objective** | | | | | | | | | | |
| LSTM-LM* | **328.0 (0.3)** | – | – | – | – | 358.1 (0.6) | – | – | – | – |
| VAE | 329.0 (0.1) | 329.1 (0.1) | 0.0 (0.0) | 0.0 (0.0) | 0.0 (0.0) | 358.3 (0.2) | 358.5 (0.2) | 0.0 (0.0) | 0.0 (0.0) | 0.0 (0.0) |
| SA-VAE | 329.2 (0.2) | 329.2 (0.2) | 0.1 (0.0) | 0.1 (0.0) | 0.8 (0.4) | 357.8 (0.2) | 357.9 (0.2) | 0.3 (0.1) | 0.3 (0.0) | 1.0 (0.0) |
| Ours | 328.2 (0.2) | 329.8 (0.2) | 5.6 (0.2) | 3.0 (0.0) | 8.0 (0.0) | **356.9 (0.2)** | 357.9 (0.2) | 3.4 (0.3) | 2.4 (0.1) | 7.4 (1.3) |

Table 6: Results on OMNIGLOT dataset. We report mean values across 5 different random restarts, and standard deviation is given in parentheses when available. For PixelCNN* we report the exact negative log likelihood.

| Model | IW | -ELBO | KL | MI | AU |
| --- | --- | --- | --- | --- | --- |
| **Previous Reports** | | | | | |
| VLAE (Chen et al., 2017) | 89.83 | – | – | – | |
| VampPrior (Tomczak & Welling, 2018) | 89.76 | – | – | – | |
| **Modified VAE Objective** | | | | | |
| VAE + anneal | 89.21 (0.04) | 89.55 (0.04) | 1.97 (0.12) | 1.79 (0.11) | 5.3 (1.0) |
| $\beta$-VAE ($\beta = 0.2$) | 105.96 (0.38) | 113.24 (0.40) | 69.62 (2.16) | 3.89 (0.03) | 32.0 (0.0) |
| $\beta$-VAE ($\beta = 0.4$) | 96.09 (0.36) | 101.16 (0.66) | 44.93 (12.17) | 3.91 (0.03) | 32.0 (0.0) |
| $\beta$-VAE ($\beta = 0.6$) | 92.14 (0.12) | 94.92 (0.47) | 25.43 (9.12) | 3.93 (0.03) | 32.0 (0.0) |
| $\beta$-VAE ($\beta = 0.8$) | 89.15 (0.04) | 90.17 (0.06) | 9.98 (0.20) | 3.84 (0.03) | 13.0 (0.7) |
| SA-VAE + anneal | **89.07 (0.06)** | 89.42 (0.06) | 3.32 (0.08) | 2.63 (0.04) | 8.6 (0.5) |
| Ours + anneal | 89.11 (0.04) | 89.62 (0.16) | 2.36 (0.15) | 2.02 (0.12) | 7.2 (1.3) |
| **Standard VAE Objective** | | | | | |
| PixelCNN* | 89.73 (0.04) | – | – | – | – |
| VAE | 89.41 (0.04) | 89.67 (0.06) | 1.51 (0.05) | 1.43 (0.07) | 3.0 (0.0) |
| SA-VAE | 89.29 (0.02) | 89.54 (0.03) | 2.55 (0.05) | 2.20 (0.03) | 4.0 (0.0) |
| Ours | **89.05 (0.05)** | 89.52 (0.03) | 2.51 (0.14) | 2.19 (0.08) | 5.6 (0.5) |

## D UNCERTAINTY OF EVALUATION

To measure the uncertainty in the evaluation stage caused by random Monte Carlo samples, we load pre-trained VAE models trained by our approach and basic VAE training, and repeat our evaluation process with 10 different random seeds. We report the mean and variance values in Table 7 and Table 8.

Table 7: Evaluation of a trained VAE model trained by our approach across 10 different random seeds. Mean values are reported and variance is given in parentheses. IW denotes the approximation to NLL we used in Section 6.

| Dataset | IW | -ELBO | KL | MI | AU |
|---|---|---|---|---|---|
| Yahoo | $327.98$ $(10^{-5})$ | $329.54$ $(5 \times 10^{-4})$ | $5.35$ $(0)$ | $3.01$ $(0.002)$ | $8$ $(0)$ |
| Yelp | $357.03$ $(10^{-5})$ | $358.25$ $(2 \times 10^{-4})$ | $3.82$ $(10^{-5})$ | $2.61$ $(0.003)$ | $8$ $(0)$ |
| OMNIGLOT | $89.03$ $(0)$ | $89.53$ $(3 \times 10^{-4})$ | $2.54$ $(0)$ | $2.21$ $(0.001)$ | $6$ $(0)$ |

Table 8: Evaluation of a trained VAE model trained by basic VAE training across 10 different random seeds. Mean values are reported and variance is given in parentheses. IW denotes the approximation to NLL we used in Section 6.

| Dataset | IW | -ELBO | KL | MI | AU |
|---|---|---|---|---|---|
| Yahoo | $328.85$ $(0)$ | $329.54$ $(1 \times 10^{-5})$ | $0.00$ $(0)$ | $0.00$ $(0)$ | $0$ $(0)$ |
| Yelp | $358.17$ $(0)$ | $358.38$ $(3 \times 10^{-5})$ | $0.00$ $(0)$ | $0.00$ $(0)$ | $0$ $(0)$ |
| OMNIGLOT | $89.41$ $(0)$ | $89.66$ $(2 \times 10^{-4})$ | $1.48$ $(0)$ | $1.39$ $(6 \times 10^{-4})$ | $3$ $(0)$ |

# E    COMPARISON WITH DIFFERENT KL-ANNEALING SCHEDULES

For the annealing baseline in Table 1 and Table 2, we implement annealing as increasing KL weight linearly from 0.1 to 1.0 in the first 10 epochs following (Kim et al., 2018), and observed posterior collapse for KL-annealing method. However, this annealing strategy may not be the optimal. In this section, we explore different KL-annealing schedules. Specifically, we increase KL weight linearly from 0.0 to 1.0 in the first $s$ iterations, and $s$ is varied as 30k, 50k, 100k, and 120k. We report results on three datasets in Table 9. The results indicate that KL-annealing does not experience posterior collapse if the annealing procedure is sufficiently slow, but it does not produce superior predictive log likelihood to our approach, which is expected because a very slow annealing schedule resembles $\beta$-VAE training in the first many epochs, and $\beta$-VAE encourages learning latent representations but might sacrifice generative modeling performance, as we already showed in Table 1 and Table 2. Also, the optimal KL annealing schedule varies with different datasets and model architectures, so that it requires careful tuning for the task at hand.

Table 9: Results on Yahoo and Yelp datasets, with different annealing schedules. Starred entries represent original annealing strategy.

| Model | Yahoo | | | | Yelp | | | | OMNIGLOT | | | |
|---|---|---|---|---|---|---|---|---|---|---|---|---|
| | NLL | KL | MI | AU | NLL | KL | MI | AU | NLL | KL | MI | AU |
| VAE + anneal (30k) | 328.4 | 0.0 | 0.0 | 0 | 357.9 | 0.2 | 0.2 | 1 | 89.18 | 2.54 | 2.19 | 10 |
| VAE + anneal (50k) | 328.3 | 0.7 | 0.7 | 4 | 357.7 | 0.3 | 0.3 | 1 | 89.15 | 3.18 | 2.58 | 10 |
| VAE + anneal (100k) | 327.5 | 4.3 | 2.6 | 12 | 356.8 | 1.9 | 1.2 | 5 | 89.27 | 4.04 | 2.97 | 16 |
| VAE + anneal (120k) | 327.5 | 7.8 | 3.2 | 18 | 356.9 | 2.7 | 1.8 | 6 | 89.32 | 4.12 | 3.00 | 15 |
| VAE + anneal* | 328.6 | 0.0 | 0.0 | 0 | 358.0 | 0.0 | 0.0 | 0 | 89.20 | 2.11 | 1.89 | 5 |
| Ours + anneal* | **326.6** | 6.7 | 3.2 | 15 | **355.9** | 3.7 | 2.3 | 10 | 89.13 | 2.53 | 2.16 | 8 |
| Ours | 328.0 | 5.4 | 3.0 | 8 | 357.0 | 3.8 | 2.6 | 8 | **89.03** | 2.54 | 2.20 | 6 |

# F    SEPARATE LEARNING RATES OF INFERENCE NETWORK AND GENERATOR

The lagging behavior of inference networks observed in Section 3 might be caused by different magnitude of gradients of encoder and decoder[9], thus another simpler possible solution to this problem is to use separate learning rates for the encoder and decoder optimization. Here we report the results of our trial by using separate learning rates. We experiment with the Yelp dataset, and keep the decoder optimization the same as discussed before, but vary the encoder learning rates to be 1x, 2x, 4x, 6x, 8x, 10x, 30x, 50x of the decoder learning rate. We notice that training becomes very unstable

---

[9]In the experiments, we did observe that the gradients of decoder is much larger than the gradients of encoder.

when the encoder learning rate is too large. Particularly it experiences KL value explosion for all the 8x, 10x, 30x, 30x, 50x settings. Therefore, in Table 10 we only report the settings where we obtained meaningful results. All of the settings suffer from posterior collapse, which means simply changing learning rates of encoders may not be sufficient to circumvent posterior collapse.

Table 10: Results on Yelp dataset varying learning rate of inference network.

| Learning Rate | NLL | KL | MI | AU |
|---|---|---|---|---|
| 1x | 358.2 | 0.0 | 0.0 | 0 |
| 2x | 358.3 | 0.0 | 0.0 | 0 |
| 4x | 358.2 | 0.0 | 0.0 | 0 |
| 6x | 390.3 | 0.0 | 0.0 | 0 |

## G   DISCUSSION ABOUT INITIALIZATION OF INFERENCE NETWORKS

In Section 3 we observe and analyze the lagging behavior of inference networks on synthetic data, but Figure 2 only shows the setting where the model is initialized at the origin. It remains unknown if a different initialization of inference networks would also suffer from posterior collapse, and whether our approach would work in that case or not. Here we explore this setting. Specifically, we add an offset to the uniform initialization we used before: we initialize all parameters as $\mathcal{U}(0.04, 0.06)$ (previously $\mathcal{U}(-0.01, 0.01)$), except the embedding weight as $\mathcal{U}(0.0, 0.2)$ (previously $\mathcal{U}(-0.1, 0.1)$). Since all parameters are positive values the output of encoder must be positive. We show the posterior mean space over course of training in Figure 6. Note that all points are located at (approximately) the same place, and are on $\mu_{\mathbf{x}, \boldsymbol{\theta}} = 0$ upon initialization, which means $\mathbf{z}$ and $\mathbf{x}$ are still nearly independent in terms of both $p_{\boldsymbol{\theta}}(\mathbf{x}|\mathbf{z})$ and $q_{\boldsymbol{\phi}}(\mathbf{z}|\mathbf{x})$. We observe that in basic VAE training these points move back to $\mu_{\mathbf{x}, \boldsymbol{\phi}} = 0$ very quickly. This suggests that the "lagging" issue might be severe only at the inference collapse state. In such a setting our approach works similarly as before.

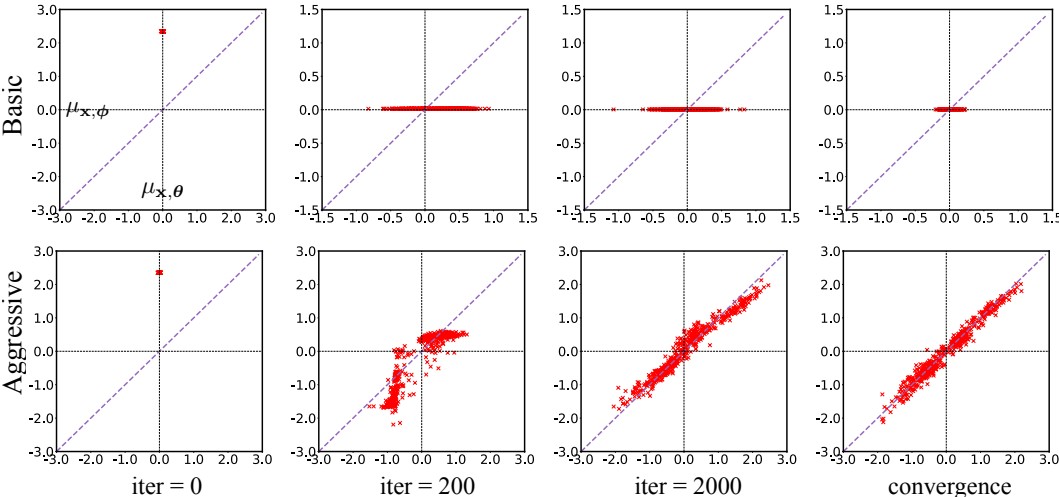

Figure 6: The projections of 500 data samples from synthetic dataset on the posterior mean space over the course of training. "iter" denotes the number of updates of generators. The top row is from the basic VAE training, the bottom row is from our aggressive inference network training. The results show that while the approximate posterior is lagging far behind the true model posterior in basic VAE training, our aggressive training approach successfully moves the points onto the diagonal line and away from inference collapse.

