# OpenReview forum: "Lagging Inference Networks and Posterior Collapse in Variational Autoencoders"
_ICLR.cc/2019/Conference_

### Official Review · AnonReviewer1 · 2018-11-01
**Neither the objective nor the model but the optimization procedure may be the key to training VAE**

**Rating:** 8
**Confidence:** 4

**Review:**

General:
The paper tackles one of the most important problems of learning VAEs, namely, the posterior collapse. Typically, this problem is attacked by either proposing a new model or modifying the objective. Interestingly, the authors considered a third option, i.e., changing the training procedure only, leaving the model and the objective untouched. Moreover, they show that in fact the modified objective (beta-VAE) could drastically harm training a VAE.

I find the idea very interesting and promising. The proposed algorithm is very easy to be applied, thus, it could be easily reproduced. I believe the paper should be presented at the ICLR 2019.

Pros:
+ The paper is written in a lucid manner. All ideas are clearly presented. I find the toy problem (Figure 2) very illuminating.
+ It might seem that the idea follows from simple if not even trivial remarks. But this impression is fully due to the fashion the authors presented their idea. I am truly impressed by the writing style of the authors.
+ I find the proposed approach very appealing because it requires changes only in the optimization procedure while the model and the objective remain the same. Moreover, the paper formalizes some intuition that could be found in other papers (e.g., (Alemi et al., 2018)).
+ The presented results are fully convincing.

Cons:
- It would be beneficial to see samples for the same latent variables to verify whether the model utilizes the latent code. Additionally, a latent space interpolation could be also presented.
- The choice of the stopping criterion seems to be rather arbitrary. Did the authors try other methods? If yes, what were they? If not, why the current stopping criterion is so unique?
- The proposed approach was applied to the case when the prior is a standard Normal. What would happen if a different prior is considered?

Neutral remark:
* Another problem, next to the posterior collapse, is the “hole problem” (see Rezende & Viola, “Taming VAEs”, 2018). A natural question is whether the proposed approach also helps to solve this issue? One possible solution to that problem is to take the aggregated posterior as the prior (e.g., (Tomczak & Welling, 2018)) or to ensure that the KL between the aggregated posterior and the prior is small. In Figure 4 it seems it is the case, however, I am really curious about the authors’ opinion on this matter.
* Can the authors relate the proposed algorithm to the wake-sleep algorithm? Obviously, the motivation is different, however, I find these two approaches a bit similar in spirit.

--REVISION--
I would like to thank the authors for their comments. In my opinion the paper is very interesting and opens new directions for further research (as discussed by the authors in their reply). I strongly believe the paper should be accepted and presented at the ICLR.

---

> ### Author Response · Authors · 2018-11-15
> **Author Response**
>
> Thanks for your encouraging comments and advice! Currently we are running additional experiments to address some of the reviewer comments. This is taking some time and we will submit a revised version once we have collected all the results. For now, we will quickly answer some of your questions.
>
> ## Q1: Latent variable interpretation
>
> We agree that providing samples would be informative. We plan to add these experiments, along with additional analysis aimed at uncovering how the latent codes are used by the generative model.
>
> ## Q2: Choice of stopping criterion
>
> In addition to the presented stopping criterion, we first tried switching back to traditional VAE training after a fixed number of epochs -- i.e. early stopping. We found that this approach can also work well, but introduces an additional hyperparameter (number of epochs) that is sensitive to datasets and model architectures. We found that stopping too early hurts the performance, and stopping too late of course hurts speed. This tradeoff needs to be tuned if epochs are specified explicitly.
>
> Intuitively, posterior collapse (and a “lagging” encoder) correspond to a lack of “dependence” between posterior samples and observed data. Based on its use in related literature, we experimented with mutual information (MI) as a simple quantitative surrogate for “dependence”. Stopping the aggressive training phase after MI stops increasing monotonically worked well in practice and avoided the need for data or model dependent tuning. Across multiple settings we found the proposed stopping criterion doesn't sacrifice performance and maintains fast training. We agree that further analysis would be interesting and suspect that similar measurements of dependence and related stopping criteria might also strike a successful balance.
>
> ## Q3: Different prior
>
> The effect of our approach under different priors would certainly be interesting to see, but is a bit beyond the scope of the current paper. We may explore this direction in future work.
>
>
> ## Q4: Hole problem
>
> Our analysis and empirical results were focused specifically on the problem of posterior collapse. We agree, however, that it would be interesting to explore how the proposed procedure (and related modifications to optimization) might affect other known issues with VAE. We hope to explore this in the future welcome any suggestions for how to do so!
>
> Regarding the “hole” problem: We were not aware of this paper, thank you for sharing it with us. Our current experimental results demonstrate that the proposed approach is able to maintain a relatively small KL(q(z) | p(z)), but this real-valued quantity is hard to interpret. We think that it is necessary to visualize the aggregated posterior and prior (or use another more direct metric) to check if the proposed approach helps solve the "hole problem".
>
> ## Q5: Connection to the wake-sleep algorithm
>
> Good point! The proposed algorithm is similar to the wake-sleep algorithm in the sense that we split encoder and decoder optimization into separate phases. Essentially, both the proposed algorithm and the wake-sleep algorithm are instances of block-coordinate ascent. The decoder update in the proposed method is analogous to the wake phase: the ELBO objective corresponds to the wake phase objective with an additional regularization term from the prior on code z. The encoder update in the proposed method is analogous to the sleep phase where decoder is fixed -- though, here, the ELBO objective is somewhat different from the sleep phase objective which aims to recover hidden code z instead of observations x.

---

### Official Review · AnonReviewer3 · 2018-11-01

**Rating:** 8
**Confidence:** 4

**Review:**

This work looks into the phenomenon of posterior collapse, and shows that training the inference network more can reduce this problem, and lead to better optima. The exposition is clear. The proposed training procedure is simple and effective. Experiments were carried out in multiple settings, though I would've liked to see more analysis. Overall, I think this is a nice contribution. I have some concerns which I hope the authors can address.

Comments:
 - I think [1] should be cited as they first mentioned Eq 5 and also performed similar analysis.
 - Were you able to form an unbiased estimate for the log of the aggregate posterior which is used extensively in this paper (e.g. MI)? Some recents works also estimate this but they use biased estimators. If your estimator is biased, please add a sentence clarifying this so readers aren't mislead.
 - Apart from KL and (biased?) MI, a metric I really would've liked to see is the number of active/inactive units as measured in [2]. I think this is a more reliable and very explainable metric for posterior collapse, whereas real-valued information-theoretic quantites can be hard to interpret.

Questions:
 - Has this approach truly completely solved posterior collapse? (e.g. can you show that the mutual information between z and x is maximal or the number of inactive units is zero?)
 - How robust is this approach to the effects of randomness during training such as initialization and use of minibatches? (e.g. can you show some standard deviations of the metrics you report in Table 1?)
 - (minor) I wasn't able to understand why the top right is optimal, as opposed to anywhere on the dashed line, in Figures 1(b) and 3?

[1] Hoffman, Matthew D., and Matthew J. Johnson. "Elbo surgery: yet another way to carve up the variational evidence lower bound."
[2] Burda, Yuri, Roger Grosse, and Ruslan Salakhutdinov. "Importance weighted autoencoders."

--REVISION--

The paper has significantly improved since the revision and I am happy to increase my score. I do still think that the claim of "preventing" or "avoiding" posterior collapse is too strong, as I agree with the authors that "it is unknown whether there is a better local optimum that [activates] more or all latent units". I would suggest not to emphasize it too strongly (ie. in the abstract) or using words like "reducing" or "mitigate" instead.

---

> ### Author Response · Authors · 2018-11-15
> **Author Response**
>
> Thank you for your comments! Your review is helpful and we are currently running additional experiments based on some of your suggestions. This will take some time and we will submit a revised version once we collect the results. For now we will quickly answer some of your questions, and describe our revision plan given your concerns.
>
>
> ## Q1: Estimator of MI
>
> The estimator we used for MI is biased because the estimator for the log of the aggregate posterior is biased. More specifically, it is a Monte Carlo estimate of an upper bound on MI. In future revisions we will be sure to provide more details and point to related work that uses the same estimate of MI. Thanks for catching the lack of detail here -- this was an oversight on our part.
>
> ## Q2: Active units
>
> Great idea! We are currently re-running experiments and keeping track of active units. In the revised version, we will include this measure for all models in Table 1.
>
>
> ## Q3: Robustness to the effects of randomness
>
> We agree that quantifying robustness to initialization is important. We are currently re-running all the models with different random seeds. Once these experiments complete, we will update the draft with mean and variance across random restarts. In our implementation, different random seeds lead to different initialization and minibatch traversal.
>
> ## Q4: Presentation suggestions
>
> (1) Thanks for pointing out this related paper [1]. We will be sure cite and include it in the discussion of related work.
>
> (2) Actually, the optimum in this cartoon might be anywhere on the dashed x=y line depending on the data and specific shape of the objective. We intended Figure 1(b) to convey that the global optimum is not located at origin, that the origin is a local optimum, and that the global optimum is somewhere on the dashed x=y line. In Figure 3 we arbitrarily chose to show a point that happens to move to top right, which must had added to the confusion. Thanks for catching this ambiguity. We will clarify the meaning of these figures in future revisions.
>
> [1] Hoffman, Matthew D., and Matthew J. Johnson. "Elbo surgery: yet another way to carve up the variational evidence lower bound."

---

> ### Author Response · Authors · 2018-11-27
> **Revision Submitted**
>
> We have completed additional experiments and submitted a revised manuscript to address the reviewer’s comments.
>
> ## Q1: Estimator of MI
>
> We have made it clear to the reader that the MI estimator is biased in Section 4.2.
>
>
> ## Q2: Active units
>
> We have computed the number of active units (AU) for all training procedures, and included AU as an additional metric in all the results tables, including the main results in Tables 1 and 2. Additionally, we plot NLL vs. AU for all training procedures in the new Figure 4.
>
>
> ## Q3: Robustness to the effects of randomness
>
> We have run all models with 5 different random seeds and report mean and standard deviations for all metrics in Table 1 and Table 2. Since outlier runs might influence mean and standard deviation a lot, we also plot each value obtained by separate runs of each method as a point in Figure 4 to visualize the uncertainties.
>
>
> ## Q4: Has this approach truly completely solved posterior collapse? (e.g. can you show that the mutual information between z and x is maximal or the number of inactive units is zero?)
>
> As shown in Tables 1 and 2, both our approach and SA-VAE behave similarly, with roughly 20-to-50% of units active. For all datasets (both language and vision), beta-VAE achieves the highest number of AU, but yields poor likelihood. Apart from beta-VAE, our approach and SA-VAE yield the highest AU in comparison with all other training approaches, and achieve the highest likelihoods overall. This matches our hypothesis since SA-VAE and our approach are the two training procedures that address inference network lag.
>
> While these results indicate that our proposed training procedure does mitigate the effects of posterior collapse, it is difficult to say whether the issue has been completely solved. For example, a large proportion of units being inactive (as we still see with our approach and SA-VAE) does not necessarily indicate posterior collapse. The objective function is highly non-convex and inactive units may be a result of other local optima that have nothing to do with collapse. This is supported by the fact that the “dying units effect” is also commonly observed in neural networks and latent variable models where posterior collapse is not a concern, e.g. [1] also observed that half of units are inactive with the IWAE objective.
>
> Further, while we agree that miniscule mutual information is an effect of posterior collapse which is a poor local optimum of the training objective, we do not think it necessarily true that the global optimum of the training objective has maximal mutual information -- or even that the model with best generalization will necessarily have maximal MI. For example, beta-VAE can have larger mutual information and more active units, but it actually fits the data poorly. During learning the model choses a subset of latent units to activate, which corresponds to some local optimum; it is unknown whether there is a better local optimum that actives more or all latent units.
>
> When looking at this as a representation learning problem, it is more intuitive that maximal MI is inherently valuable -- but, in this paper, we primarily view VAE as a probabilistic model for which good generalization to unseen data is of highest concern.
>
> [1] Burda, Yuri, Roger Grosse, and Ruslan Salakhutdinov. "Importance weighted Autoencoders."

---

### Official Review · AnonReviewer2 · 2018-11-02
**Reasonable solution to posterior collapse but needs uncertainty quantification and more effort on baselines and debunking alternative explanations**

**Rating:** 7
**Confidence:** 4

**Review:**

Response to Authors
-------------
I've read all other reviews and the author responses. Most responses to my issues seem to be "we will run more experiments", so my review scores haven't changed. I'm glad the authors are planning many revised experiments, and I understand that these take time. It's too bad revised results won't be available before the review revision deadline (tomorrow 11/26). I guess I'm willing to take the author's promises to update in good faith. Thus, I think this is an "accept", but only if the authors really do follow through on promises to add uncertainty quantification and include some complete comparisons to KL annealing strategies.

Review Summary
--------------

Overall, I think the paper offers a reasonable story for why its proposed innovation -- an alternative scheduling of parameter-specific updates where encoder parameters are always trained to convergence during early iterations -- might offer a reliable way to avoid posterior collapse that is far faster and easier-to-implement than other options that require some per-example iterations (e.g. semi-amortized VAE). My biggest concerns are that relative performance gains (in bound quality) over alternatives are not too large and hard to judge as significant because no uncertainty in these estimates is quantified. Additionally, I'd like to see more careful evaluation of the KL annealing baseline and more attention to within-model comparisons (do you really need to update until convergence?).

Given the method's simplicity and speed, I think with a satisfactory rebuttal and plan for revision I would lean towards acceptance.

Paper Summary
-------------
The paper investigates a common problem known as "posterior collapse" observed when training generative models such as VAEs (Kingma & Welling 2014) with high-capacity neural networks. Posterior collapse occurs when the encoder distribution q(z|x) (parameterized by a NN) becomes indistinguishable from the generative prior on codes p(z), which is often a local optima of the VI ELBO objective. While other better fixed points exist, once this one is reached during optimization it is hard to escape using the typical local gradient steps for VAEs that jointly update the parameters of an encoder and a decoder with each gradient step.

The proposed solution (presented in Alg. 1) is to avoid joint gradient updates early in training, and instead use an alternating update scheme where after each single-gradient-step decoder parameter update, the encoder is updated with as many gradient steps as are needed to reach convergence. This proposed scheme, which the paper terms "aggressive updates", forces the encoder to better approximate the true posterior p(z|x) at each step.

Experiments study a synthetic task where visualizing the evolution of true posterior mean of p(z|x) side-by-side with approximate q(z|x) is possible in 2D, as well as benchmark comparisons to several other methods that address posterior collapse on text modeling (Yahoo, Yelp15) and image modeling (Omniglot). Studied baselines include annealing the KL term in the VI objective, the \beta VAE (which keeps the KL term fixed with a weight \beta), and semi-amortized VAEs (SA-VAEs, Kim et al. 2018). The presented approach is said to reach better values of the log likelihood while also being ~10x faster to train than the Kim et al. approach on large datasets.

Significance and Originality
----------------------------
There exists strong interest in deploying amortized VI to fit sophisticated models efficiently while avoiding posterior collapse, so the topic is definitely relevant to ICLR. Certainly solutions to this issue are welcome, though I worry with the crowded field that performance is starting to saturate and it is becoming hard to identify significant vs. marginal contributions. Thus it's important to interpret results across multiple axes (e.g. speed and heldout likelihood).

The paper does a nice job of highlighting related work on this problem, and I'd rate its methodological contributions as clearly distinct from prior work, even though the eventual procedure is simple.

The closest related works in my view are:

* Krishnan et al. AISTATS 2018, where VAE joint-training algorithms for nonlinear factor analysis problems are shown to be improved by an algorithm that uses the encoder NN as an *initialization* and then doing several standard SVI updates to refine per-example parameters. Encoder parameters are updated via gradient updates, *after* the decoder parameters are updated (not jointly).

* SA-VAEs (Kim et al. ICML 2018) which studies VAEs for deep text models and develops an algorithm that at each a new batch uses the encoder to initialize per-example parameters, updates these via several iterations of SVI, then *backpropagates* through those updates to compute a gradient update of the encoder NN.

Compared to these, the detailed algorithm presented in this work is both distinct and simpler. It does not require any per-example parameter updates, instead it only requires a different scheduling of when encoder and decoder NN updates occur.


Concerns about Technical Quality (prioritized)
----------------------------------------------

## C1: Without error bars in Table 1 and 3, hard to know which gaps are significant

Are 500 Monte Carlo samples enough to be sure that the numbers reported in Table 1 are precise estimates and not too noisy? How much error is there in the estimation of various quantities like the NLL or the KL if we repeated 500-MC samples 5x or 10x or 25x? My experience is that even with 100 or more samples, evaluations of the ELBO bound for classic VAEs can differ non-trivally. I'd like to see evidence that these quantities are estimated with certainty, or (even better) some direct reporting of the uncertainties across several estimates.


## C2: Baseline comparison to KL annealing needs to be more thorough

The current paper dismisses the strategy that annealing the KL term as ineffective in addressing posterior collapse (e.g. VAE + anneal has a 0.0 KL term in Table 1). However, it's not clear that a reasonable annealing schedule was used, or even that any reasonable effort was made to try more than one schedule. For example, if we set the KL term to exactly 0.0 weight, the optimization has no incentive to push q towards the prior, and thus posterior collapse *cannot* occur. It may be that this leads to other problems, but it's unclear to me why a schedule that keeps the KL term weight exactly at 0 for a few updates and then gradually increases the weight should lead to collapse. To me, the KL annealing story is much simpler than the presented approach and I think as a community we should invest in giving it a fair shot. If the answer is that annealing takes too long or the schedule is tough to tune, that's sensible, but I think the claim that annealing still leads to collapse just means the schedule probably wasn't set right.

Notice that "Ours" is improved by "Ours+Annealing" for 2 datasets in Table 1. So annealing *can* be effective. Krishnan et al. 2018's Supplementary Fig. 10 suggests that if annealing is slow enough (unfolding over 100000 updates instead of 10000 updates), then KL annealing will get close to pure SVI in effective, non-collapsed posterior approximation. The present paper's Sec. B.3 indicates that the attempted annealing schedule was 0.1 to 1.0 linearly over 10 epochs with batch size 32 and train set size 100k, which sounds like only 30k updates of annealing were performed. I'd suggest comparing against KL annealing that both starts with a smaller weight (perhaps exactly at 0.0) and grows much slower.


## C3: Results do not analyze variability due to random initialization or random minibatch traversal

Many factors can impact the final performance values of a model trained via VI, including the random initialization of its parameters and the random order of minibatches used during gradient updates. Due to local optima, often best practice is to take the best of many separate initializations (see several figures in Bishop's PRML textbook). The present paper doesn't make clear whether it's reporting single runs or the best of many runs. I suggest a revision is needed to clarify. Quantifying robustness to initialization is important.


## C4: Results do not analyze relative sensitivity of encoder and decoder to using the same learning rate

One possible explanation for "lagging" might be that the gradient vectors of the encoder and the decoder have different magnitudes, and thus using the same fixed learning rate for both (as seems to be done from a skim of Sec. B) might not be optimal. Perhaps a quick experiment that separately tunes learning rates of encoder and decoder is necessary? If the learning rate for encoder is too small, this could easily explain the lagging when using joint updates.


## C5: Is it necessary to update until convergence? Or would a fixed budget of 25 or 100 updates to the encoder suffice?

In Alg. 1, during the "aggressive" phase the encoder is updated until convergence. I'd like to see some coverage of how long this typically takes (10 updates? 100 updates?). I'd also like to know if there are significant time-savings to be had by not going *all* the way to convergence. It's concerning that in Fig. 1 convergence on a toy dataset takes more than 2000 iterations.


## C6: Sensitivity to the initialization of the encoder is not discussed and could matter

In the synthetic example figure, it seems the encoder is initialized so that across many examples, the typical encoding will be near the origin and thus favored under the prior. Thus, the *initialization* is in some ways setting optimization up for posterior collapse. I wonder if some more diverse initialization might avoid the problem.



Presentation comments
---------------------

Overall the paper reads reasonably. I'd suggest mentioning the KL annealing comparison a bit earlier, but otherwise I have few complaints.

I'm not sure I like the chosen terminology of "aggressive" update. The procedure is more accurately a "repeat-until-convergence" update. There's nothing aggressive about it, it's just repeated.


Line-by-line Detailed comments
------------------------------

Citations for "traditional" VI with per-example parameters should go much further back than 2013. For example, Matthew Beal's thesis, work by Blei in 2003 on LDA, or work by MacKay or M.I. Jordan or others even further back.

Alg 1 Line 12: This update should be to \theta (model parameters), not \phi (approx posterior parameters).

Alg 1: Might consider using notation like g_\theta to denote the grad. of specific parameters, rather than have the same symbol "g" overloaded as the gradient of \theta, \phi, and both in the same Algo.


Fig. 3: This is interesting, but I think it's missing something as a visualization of the algorithm. There's nothing obvious visually that indicates the encoder update involves *many* steps, but the decoder update is only one step. I'd suggest at least turning each vertical arrow into *many* short arrows stacked end-to-end, indicating many steps. Also use a different color (not green for both).

Fig. 4: Shows various quantities like KL(q, prior) traced over optimization. This figure would be more illuminating if it also showed the complete ELBO objective and the expected log likelihood term. Then it would be clear why annealing is failing to avoid posterior collapse.

Table 1: How exactly is the negative log likelihood (NLL) computed? Is it the expected value of the data likelihood: -1 * E_q[log p(x|z)]? Or is it the variational lower bound on marginal likelihood?

---

> ### Author Response · Authors · 2018-11-15
> **Author Response [2/2]**
>
> ## Q4: Separate learning rates of encoder and decoder
>
> This is a good point! When we first observed the "lagging" behaviour we also found that the gradient of the encoder and decoder had very different magnitudes. We tried doing exactly what you propose: tuning the learning rates for the encoder and decoder separately, as well as experimenting with alternative optimization methods as potential solutions -- but nothing worked. We realize that readers might be curious about this matter, thus we will include further discussion in the paper and additional negative experimental results as support.
>
>
> ## Q5: Is it necessary to update until convergence ?
>
> This is a good question! In practice, of course, we never reach *exact* convergence, thus the question is really about how close to convergence is required in the inner loop update. In our current implementation, we break the inner loop when the ELBO objective stays the same or decreases across 10 iterations. Note that we don't perform separate learning rate decay in the inner loop so this convergence condition is not strict, but empirically we found it to be sufficient. Across all four datasets (on synthetic, Yahoo, Yelp, and OMNIGLOT) in practice this yields roughly 30 - 100 updates per inner loop update. We also want to clarify that Fig.2 doesn't imply our approach takes 2000 updates to converge in one single inner loop, the notation "iter" in Figure 2 represents outer loop iterations instead of inner loop iterations (we will clarify this in future revisions -- thank you for pointing out the ambiguity).
>
> In preliminary experiments we tried using a fixed budget of encoder updates, similar to the approach you suggest. While not reported in the current revision, our takeaway from these experiments was the following: (1) Generally speaking, the final model fit is better when the encoder update is near convergence. (2) Performing a sufficient number of updates above some threshold in the inner loop is *critical* for avoiding posterior collapse -- we found that this "sufficient number" is sensitive to dataset and model architecture. (3) We found, empirically, that the minimal fixed budget of inner loop iterations required to avoid posterior collapse was not meaningfully smaller than the number of updates resulting from our proposed approach and implementation. Therefore, we concluded that the fixed budget approach would not lead to worthwhile speedups in practice, and that our simpler proposed approach represents a good tradeoff between performance and speed. We will include this discussion in future revisions.
>
> ## Q6: Initialization of encoder
>
> We had also considered whether a different initialization for the encoder might help avoid posterior collapse, but did not conduct experiments to test this hypothesis. Considering your concern, we plan to at least conduct experiments where we initialize all the encoder parameters to positive values (so that the approximate posterior mean is not located at origin upon initialization). We will discuss this point in future revisions and include experimental results if they are interesting.

---

> ### Author Response · Authors · 2018-11-15
> **Author Response [1/2]**
>
> We appreciate your thorough review and detailed comments! Your suggestions will be helpful in improving the paper. We are currently running additional experiments to address some of your questions and comments. This is taking some time and we will submit a revised version once we collect all the results. For now, we will quickly answer some questions, and describe our revision plan to address your concerns.
>
> ## Q1: Uncertainty quantification for random initialization or random minibatch traversal
>
> The reported results in the submitted paper are from single runs. We agree that measuring robustness to initialization is important. We are currently re-running all the models with multiple random seeds. After these experiments finish, we will report the mean and variance across different runs. In our implementation, different random seeds lead to different initialization and minibatch traversal.
>
>
> ## Q2: NLL approximation and its uncertainty quantification
>
> We approximated log likelihood with 500 importance weighted samples as in [1], which does yield (an Monte Carlo estimate of) a lower bound on marginal likelihood. We will revise to make this more clear to readers. This lower bound is tighter than ELBO as shown in [1] (we also reported both NLL and ELBO values in Appendix C). To measure the uncertainty in these evaluation metrics due to their Monte Carlo estimates, in the revised version we report variance from repeating evaluation multiple times on each trained model.
>
> [1] Burda, Yuri, Roger Grosse, and Ruslan Salakhutdinov. "Importance weighted autoencoders."
>
>
> ## Q3: Baseline comparison to KL annealing
>
> We agree that a more thorough comparison with KL annealing should be included. Past experience with unsuccessful attempts at KL annealing on several practical problems was actually one motivation for the current work. However, in many cases KL annealing does work well when tuned properly. We will be sure to include a more complete comparison with various KL annealing strategies in the revised version -- we have some practical experience here and will describe the tuning procedures in detail in revision. It is worth noting that one strength of the proposed approach in comparison with KL annealing is that it requires far less tuning in practice because it has fewer hyperparameters than annealing strategies do.

---

> > ### Comment · AnonReviewer2 · 2018-11-25
> > **Suggestion for uncertainty quantification**
> >
> > RE quantifying the uncertainty of performance metrics across different runs of different methods: I would suggest that with small sample size there are better ways to report an estimate and its uncertainty than just the mean and standard deviation. If you have any one outlier run, it can impact both measures a lot. Also, reporting std. dev. assumes symmetric errors, but the asymmetry of errors could be important.
> >
> > Recommended way: Graphically in a figure (not a table). Show each value obtained by separate runs of each method as a point. Readers can compare visually the spread within-runs and between-runs to make a judgement about ranking of methods.
> >
> > Second best way: Table with medians and min/max (or low/high percentile). These values should be less sensitive to outliers than mean/std dev.

---

> > > ### Author Response · Authors · 2018-11-25
> > > **Author Response**
> > >
> > > Thank you for your update and experiments suggestion! We followed our promises earlier and all the planned experiments are almost ready now. We will make sure to submit the revised version before the revision ddl.

---

> > ### Comment · AnonReviewer2 · 2018-11-25
> > **Idea for a simple "adaptive" KL annealing procedure**
> >
> > I agree that if we force KL annealing to pick only one annealing schedule up front, the presented approach is perhaps simpler because you don't need to pick the tuning parameters of the schedule or try many different values (though should clarify what hyperparameter choices are implicit in the test for convergence in the presented approach).
> >
> > However, I think there's a simple "adaptive" KL strategy that could be explored:
> > * start with KL weight equal to 0.0
> > * gradually increase at linear rate to KL weight of 1.0 over X iterations
> > * if during the run any posterior collapse is detected, immediately reset the weight to 0.0 and make the rate of increase Y times slower
> >
> > I'd think this schedule has to work eventually, and it shouldn't be too hard to find reasonable values of X and Y.

---

> ### Author Response · Authors · 2018-11-27
> **Revision Submitted**
>
> We have submitted the revised manuscript and made the following modifications to address the reviewer’s comments.
>
> ## Q1: Uncertainty quantification for random initialization or random minibatch traversal
>
> We have run all methods with 5 different random seeds and report mean and standard deviations for all metrics in Table 1 and Table 2. Following the reviewer’s suggestion we also plotted each value obtained by separate runs of each method as a point in Figure 4 to visualize the uncertainties.
>
> ## Q2: Uncertainty quantification of NLL approximation
>
> We repeated the evaluation process 10 times with different random seeds, and report variance in Appendix D (Table 7 and Table 8). For a trained VAE model the variance of our NLL estimation is smaller than 0.001 on all datasets.
>
>
> ## Q3: Baseline comparison to KL annealing
>
> We started KL weight from exactly 0.0 and linearly increased it to 1.0 in the first X iterations. We tried different X and report the results in Appendix E (Table 9). KL-annealing method does not experience posterior collapse if the annealing procedure is sufficiently slow, but it does not produce superior predictive log likelihood to our approach.
>
>
> ## Q4: Separate learning rates of encoder and decoder
>
> We varied the learning rate of encoder and discussed the results in Appendix F (Table 10).
>
>
> ## Q5: Is it necessary to update until convergence ?
>
> As a follow-up to our earlier response to this question, we experiment with a fixed budget of encoder updates and report results in Section 6.5 (Table 4).
>
>
> ## Q6: Initialization of encoder
>
> We explored the setting where the model is not initialized at origin, and discussed it in Appendix G (Figure 6).

---

### Public Comment · ~Yoon_Kim1 · 2018-10-11
**a very nice paper that proposes a simple solution to address posterior collapse**

Hi, thanks for this great paper! Addressing posterior collapse in VAEs is an important issue in the field.

I particularly liked the breakdown of posterior collapse into two failure modes: inference collapse (where KL(p(z | x), p(z)) > 0 but KL(q(z | x), p(z)) = 0) and model collapse (where KL(p(z | x), p(z)) = 0). This paper shows that inference collapse happens first during optimization, and proposes a simple yet robust way to mitigate this (update the inference network more aggressively when collapse is happening).

I had a small question: for Figure 2, I wonder if it is possible to directly estimate the KL's instead of the means?
i.e. replace the vertical axis with KL(q(z | x), p(z))  and the horizontal axis with KL(p(z | x), p(z)). I could be wrong, but it seems like KL(p(z | x), p(z))  should be estimable by obtaining samples from p(z|x) with MCMC and calculating

1/M \sum_{m=1}^M log [p(z_m | x)/p(z_m)] = 1/M \sum_{m=1}^M log [p(x|z_m) / p(x)],

where z_m are samples from p(z|x) and p(x) is estimated with importance samples (from the prior). This estimator would be biased but seems like it would converge a.s. to KL(p(z|x), p(z)) under mild conditions. I would image the plots would remain roughly unchanged, but this might generalize existing plots to consider more than just the first moments.

---

> ### Author Response · Authors · 2018-10-12
> **Thanks for your comments and advice, and the KL plots may not be a better choice than the first moments plots**
>
> Thanks for your encouraging comments and advice !
>
> I think you are right that KL(p(z|x), p(z)) is estimable with the sampling method you gave. Also, we agree that plotting with KL terms is able to generalize the plots to consider more than the first moments and even higher dimensions of latent variables.
>
> However, I think the KL value plots miss to include the distance information between q(z|x) and p(z|x), which is crucial for the analysis in this paper. Two distributions that have similar KL divergence to prior might have very different moments. This distance information is important to convey in posterior space plots since we are emphasizing LAGGING inference distribution compared with true model posterior and the aggressive training of inference net is to make q(z|x) and p(z|x) closer. We cannot really say that q(z|x) and p(z|x) is close given KL(q(z|x), p(z)) and KL(p(z|x), p(z)) are close, which makes the diagonal line in these plots meaningless.
>
> Through Figure 2 and Figure 3 we want to show the moving trajectory of q(z|x) and p(z|x), not only their relationship with prior p(z) (which KL plots can reflect), but also the relationship between themselves (which KL plots cannot reflect). Due to challenge of accurate visualization, we compromized to characterize distribution with the first moments, which we believe is a reasonable approximation given the plots and quantitative results on real dataset in experiments.
>
> It might be worth visualizing both the first moments and KL values. I guess the plots for basic VAE and our approach might remain roughly unchanged, but the plots for other regularization methods like beta-VAE may be very different (we didn’t show this in the paper though). We will consider adding  KL plots in future revisions.

---

> > ### Public Comment · ~Yoon_Kim1 · 2018-10-12
> > **good point!**
> >
> > Ah yes, that's a good point. The diagonals in Figure 2 are certainly nice to see :). Thanks for the quick answer!
> >
> > (Maybe another way to visualize would be to plot KL(q(z | x), p(z)), KL(q(z | x), p(z |x)), and KL(p(z|x), p(z)) averaged over some batch of x's as training progresses. But it seems like Figure 2 shows the phenomenon pretty clearly regardless)

---

### Public Comment · ~Artem_Sobolev1 · 2018-11-06
**Could the variance of the inference network gradients be a problem?**

In "Sticking the Landing: Simple, Lower-Variance Gradient Estimators for Variational Inference" [1] it was shown that a naive differentiation of the ELBO w.r.t. φ (q's parameters) leads to a ∇ log q(z) term which has zero expectation, but contributes significant variance to the gradient estimate. Could the lagging you observed be explained by high variance of the gradients?

[1]: https://arxiv.org/abs/1703.09194

---

> ### Author Response · Authors · 2018-11-14
> **Author Response**
>
> This is a good point, and thanks for sharing the paper with us. In practice we found that using more Monte Carlo samples in our training algorithm helps improve the performance (this is expected because more Monte Carlo samples lead to low-variance gradient estimator that makes the inner-loop encoder optimization better), but using more Monte Carlo samples in standard VAE training is not sufficient to mitigate "lagging", which means lagging may not be explained by the high variance of gradients alone.
>
> While we do think that high variances of the gradients might partially contribute to the lagging of encoders, at this stage we don't have experimental evidence about if a better gradient estimator (as you mentioned) would suffice to address this problem.

---

### Author Response · Authors · 2018-11-27
**Revision Submitted**

We have submitted a revised manuscript and made the following modifications to address the reviewers' major concerns:

-- Included number of active units as an additional metric in all results tables
-- Reported mean and standard deviation across different random seeds in the main results tables
-- Added Figure 4 to show each value obtained by separate runs of each method to visualize uncertainties
-- Added more thorough comparison with KL annealing baseline (see Appendix E, Table 9)
-- Added experiment to discuss separate learning rates of encoder and decoder (Appendix F, Table 10)
-- Added results with a fixed budget of encoder updates (see Section 6.5, Table 4)
-- Added experiment to explore the setting where the model is not initialized at origin (see Appendix G, Figure 6).

While limited by time in the response period, we do still plan to address *all* the reviewer’s comments including discussion of the stopping criterion and latent variable interpretation in future revisions. We also welcome any further feedbacks to improve this paper !

---

### Public Comment · ~Jaemin_Cho1 · 2018-12-21
**Missing reference for posterior collapse mitigation**

Nice work & Congrats for acceptance!
I would like to point our work which also mitigates posterior collapse :)
https://arxiv.org/abs/1804.03424

---

> ### Author Response · Authors · 2019-01-27
> **Reply**
>
> Thanks for pointing out this related work, we have cited it appropriately.

---

### Meta-Review · Area_Chair1 · 2018-12-13

**Confidence:** 5
**Recommendation:** Accept (Poster)

**Metareview:**

This paper introduces a method that aims to solve the problem of 'posterior collapse' in variational autoencoders (VAEs). The problem of posterior collapse is well-documented in the VAE literature, and various solutions have been proposed. Existing proposed solutions, however, aim to solve the problem by either changing the objective function (e.g. beta-VAE) or by changing the prior and/or approximate posterior models. The proposed method, in contrast, aims to solve the problem by bringing the VAE optimization procedure closer to the EM optimization procedure. Every iteration in optimization consists of SGD updates to the inference model (E-step), performed until the approximate posterior converges. This is followed by a single SGD update of the generative model. The multi-update E-step makes sure that the M-step optimizes something closer to the marginal log-likelihood, compared to what we would normaly do in VAEs (joint optimization of both inference model and generative model).

The experiments are relatively small-scale, but convincing.

The reviewers agree that the method is clearly described, and that the proposed technique is well supported by the experiments. We think that this work will probably be of high interest to the ICLR community.